# Enhancing Affine Maximizer Auctions with Correlation-Aware Payment

Haoran Sun [1]   Xuanzhi Xia [2]   Xu Chu [1]   Xiaotie Deng [1]

## Abstract

Affine Maximizer Auctions (AMAs), a generalized mechanism family from VCG, are widely used in automated mechanism design due to their inherent dominant-strategy incentive compatibility (DSIC) and individual rationality (IR). However, as the payment form is fixed, AMA's expressiveness is restricted, especially in distributions where bidders' valuations are correlated. In this paper, we propose Correlation-Aware AMA (CA-AMA), a novel framework that augments AMA with a new correlation-aware payment. We show that any CA-AMA preserves the DSIC property and formalize finding optimal CA-AMA as a constraint optimization problem subject to the IR constraint. Then, we theoretically characterize scenarios where classic AMAs can perform arbitrarily poorly compared to the optimal revenue, while the CA-AMA can reach the optimal revenue. For optimizing CA-AMA, we design a practical two-stage training algorithm. We derive that the target function's continuity and the generalization bound on the degree of deviation from strict IR. Finally, extensive experiments showcase that our algorithm can find an approximate optimal CA-AMA in various distributions with improved revenue and a low degree of violation of IR.

## 1. Introduction

Recently, differentiable economics (Shen et al., 2019; Dütting et al., 2023; Curry et al., 2023; Wang et al., 2024) has attracted significant attention within automated mechanism design as it can discover auctions that demonstrate superior empirical performance. In revenue-maximizing auction design, existing methods are broadly categorized into two classes: (1) *characterization-free* methods, which

[1]CFCS, School of Computer Science, Peking University [2]Department of Computer Science and Technology, Tsinghua University. Correspondence to: Xu Chu <chu_xu@pku.edu.cn>, Xiaotie Deng <xiaotie@pku.edu.cn>.

*Proceedings of the $43^{rd}$ International Conference on Machine Learning*, Seoul, South Korea. PMLR 306, 2026. Copyright 2026 by the author(s).

directly employ neural networks to approximate auction mechanisms (Rahme et al., 2021b; Peri et al., 2021; Duan et al., 2022; Ivanov et al., 2022; Dütting et al., 2023), and (2) *characterization-based* methods, which optimize within structured mechanism families with well-defined game-theoretic properties (Shen et al., 2019; Curry et al., 2023; Duan et al., 2023; 2026; Wang et al., 2024). Among the later, Affine Maximizer Auctions (AMAs), a family of mechanisms extended from Vickery-Clarke-Groves (VCG) (Vickrey, 1961; Karp, 2010), are particularly notable for inherently guaranteeing dominant-strategy incentive compatibility (DSIC), individual rationality (IR), and preventing overallocation. Recent work on optimizing AMAs has demonstrated strong empirical performance and computational efficiency (Curry et al., 2023; Duan et al., 2023; 2026).

However, prior AMA-based methods have primarily focused on evaluations under bidder-independent distributions, where the expressive limitations of VCG-style payment rules may not be fully apparent. In certain bidder-correlated settings, this VCG-style payment rule exhibits a critical constraint: *a bidder's payment can only be a non-decreasing function of other bidders' valuations*. This inherent limitation significantly reduces their payment flexibility compared to characterization-free approaches (Dütting et al., 2023) or methods that use more expressive mechanism families, such as menu-based mechanisms (Shen et al., 2019; Wang et al., 2024). To illustrate this limitation, we present a simple example where AMAs fail to achieve optimal revenue.

**Example 1.1.** *Consider a single-item auction with two bidders where valuations are perfectly negatively correlated ($v_1 = 1 - v_2$) and each marginal valuation is drawn uniformly from $[0, 1]$.*

In this setting, the optimal DSIC and IR mechanism extracts the full surplus by setting a personalized reserve price for bidder 1 of $1 - v_2$ and for bidder 2 of $1 - v_1$. In this auction, bidder 1 wins if $v_1 > 0.5$ and pays $1 - v_2$, which is a decreasing function of $v_2$. However, in the AMA framework, the payment is structured differently. Let $\lambda_0, \lambda_1, \lambda_2$ be the boost variables for allocating the item to no one, bidder 1, and bidder 2, respectively, and let the weights $w_1 = w_2 = 1$ (as two bidders are symmetric). When bidder 1 wins, their payment is $p_1 = \max\{v_2 + \lambda_2, \lambda_1, \lambda_0\} - \lambda_1$. As this payment is necessarily a non-decreasing function of $v_2$,

no AMA can replicate the optimal mechanism. Although introducing randomness can mitigate this limitation (Curry et al., 2023; Duan et al., 2023), it also creates a non-zero probability of reserving the item, which results in revenue loss.

Motivated by this, we aim to enhance AMA's expressiveness in bidder-correlated settings with minimal modification. In this study, we introduce the Correlation-Aware Affine Maximizer Auction (CA-AMA), which incorporates an additional correlation-aware payment term, $p_i^{\text{Cor}}$, for each bidder. By setting $p_i^{\text{Cor}}$ to be independent of bidder $i$'s bid, CA-AMA inherently preserves the DSIC property. We formalize the problem of identifying the optimal CA-AMA as an optimization problem subject to IR constraints. Theoretically, we demonstrate that in single-item auctions under certain distributions, CA-AMA can achieve optimal revenue where standard AMAs perform arbitrarily poorly. We then propose a two-stage training algorithm for optimizing CA-AMA, whose feasibility is supported by the continuity of the target function and a generalization bound on the degree of IR violation. Finally, we conduct extensive experiments across various distributions in single-item and multi-item auctions. The results demonstrate that our algorithm effectively finds an approximately IR mechanism that achieves significantly improved revenue compared to standard AMAs.

The remainder of this paper is organized as follows. Section 2 first introduces the necessary preliminaries. Section 3 then demonstrates the limitations of standard AMAs and proposes the CA-AMA framework. Following this, Section 4 details the optimization of CA-AMA. We present experimental results in Section 5 and conclude the paper in Section 6.

## 2. Preliminary

We consider the sealed-bid auction with $n$ bidders $[n] = \{1, 2, \ldots, n\}$ and $m$ items $[m] = \{1, 2, \ldots, m\}$. Each bidder $i$ has a private valuation on all item combinations, denoted by $\boldsymbol{v}_i = (v_{is})_{s \subseteq [m]}$, where $v_{is}$ is the bidder's valuation of an item combination $s \subseteq [m]$. We mainly consider the *additive* valuation, *i.e.*, $v_{is} = \sum_{j \in s} v_{ij}$ for all $i \in [n]$ and $s \subseteq [m]$. So a bidder's valuation is expressed by $\boldsymbol{v}_i = (v_{ij})_{j \in [m]}$.

A valuation profile $V = (\boldsymbol{v}_1, \boldsymbol{v}_2, \ldots, \boldsymbol{v}_n)$ is a collection of all bidders' valuations. We assume that $V$ has an underlying distribution $\mathcal{F}$ and the support is bounded, $\text{supp}(F) \subseteq [0, 1]^{n \times m}$. In an auction, each bidder $i$ reports a bid $\boldsymbol{b}_i$, which does not necessarily equal its real valuation $\boldsymbol{v}_i$. The auctioneer does not know the true valuation profile $V$ nor the distribution $\mathcal{F}$ but can observe the bidding profile $B = (\boldsymbol{b}_1, \boldsymbol{b}_2, \ldots, \boldsymbol{b}_n)$. We use $V_{-i} =$

$(\boldsymbol{v}_1, \ldots, \boldsymbol{v}_{i-1}, \boldsymbol{v}_{i+1}, \ldots, \boldsymbol{v}_n)$ to represent the valuation profile except for bidder $i$, and $B_{-i}$ with the similar meaning. The marginal distribution is represented by $\mathcal{F}_i(V_{-i})$ for bidder $i$'s valuation. When the bidders are *independent*, this marginal distribution does not depend on $V_{-i}$, which means that $\mathcal{F}_i(V_{-i}) \equiv \mathcal{F}_i$ for any $V_{-i}$. When the bidders' valuation distributions are *correlated*, such a relationship does not hold.

### 2.1. Revenue-Maximizing Auction Design

An auction mechanism $(g, p)$ consists of an allocation rule $g$ and a payment rule $p$. For a given bidding profile $B$, $g(B) \subseteq [0, 1]^{n \times m}$ is the allocation matrix. The allocation rule has to satisfies that $\sum_{i=1}^{n} g(B)_{ij} \leq 1$ for any $j \in [m]$. If the mechanism is *deterministic*, we further restrict that the allocation matrix $g(B)_{ij} \subseteq \{0, 1\}$ for all $i$ and $j$. The payment rule $p_i(B) \geq 0$ determines the value the bidder $i$ has to pay. Following the literature (Duan et al., 2023; Dütting et al., 2023), we assume that the bidders are utility maximizers and have quasi-linear utility. For a mechanism $(g, p)$, the utility of bidder $i$ with true valuation $\boldsymbol{v}_i$ when the bid profile is $B$ can be written as $u_i(\boldsymbol{v}_i, B; g, p) :=$ $\boldsymbol{v}_i \cdot g(B)_i - p_i(B)$. If the mechanism $(g, p)$ which we are referring to does not raise ambiguity, we will use $u_i(\boldsymbol{v}_i, B)$ for simplicity.

The auction mechanism will be announced publicly at first, so bidders can statically report their valuation to gain a higher utility. We consider the following properties from classic auction theory (Myerson, 1981).

**Definition 2.1** (DSIC). A mechanism $(g, p)$ satisfies *dominant-strategy incentive compatibility* if for $\forall i$, $B_{-i}$, $\boldsymbol{v}_i$, and $\boldsymbol{b}_i$, we have $u_i(\boldsymbol{v}_i, (\boldsymbol{v}_i, B_{-i})) \geq u_i(\boldsymbol{v}_i, (\boldsymbol{b}_i, B_{-i}))$.

**Definition 2.2** (IR). A mechanism $(g, p)$ satisfies *individual rationality* if for $\forall i$, $(\boldsymbol{v}_i, V_{-i}) \in \text{supp}(\mathcal{F})$, we have $u_i(\boldsymbol{v}_i, (\boldsymbol{v}_i, V_{-i})) \geq 0$.

Note that the definition is slightly different from the *ex-post IR*, which requires $u_i(\boldsymbol{v}_i, (\boldsymbol{v}_i, B_{-i})) \geq 0$ for all $i$, $\boldsymbol{v}_i$ and $B_{-i}$. This is weaker than ex-post IR but stronger than ex-interim IR, as it requires the utility to be non-negative on each point $(\boldsymbol{v}_i, V_{-i})$ that can be realized by $\mathcal{F}$. We adopt this definition to facilitate theoretical analysis; since our mechanism is DSIC, it is reasonable to assume truthful reporting and thus exclude valuation profiles that will never be realized. In practice, we also propose an intuitive method to extend CA-AMA to satisfy ex-post IR, which is discussed in Section 4 and validated in our experiments.

*Remark* 2.3 (Motivation for IR Relaxation). We clarify that **addressing the revenue limitations in value-correlated scenarios was our primary motivation**, and the concept of "IR regret" emerged *a posteriori* strictly as a technical requirement for practical neural network optimization. Our theoretical formulation (CA-AMA-OPT in Section 3.2) tar-

gets strict IR constraints; the regret penalty is merely an operational tool to guide gradient-based training, as we cannot enforce hard IR constraints at every optimization step.

The optimal auction design is to find the revenue-maximizing DSIC and IR auction mechanism under a certain distribution $\mathcal{F}$, which can be formulated as the following optimization problem.

$$\max_{g,p} \quad \text{REV}_{\mathcal{F}} := \mathbb{E}_{V \sim \mathcal{F}} \sum_{i=1}^{n} p_i(V) \tag{OPT}$$
$$\text{s.t.} \quad \text{Mechanism } (g, p) \text{ satisfies DSIC and IR.}$$

### 2.2. Affine Maximizer Auctions

AMAs are a family of auction mechanisms generalized from the VCG (Vickrey, 1961; Karp, 2010) auction. An AMA can be parameterized by $(\mathcal{A}, \boldsymbol{w}, \boldsymbol{\lambda})$. $\mathcal{A} = \{A_1, \cdots, A_S\}$ is a set of $S$ distinct candidate allocations, $w_i > 0$ is the weight for bidder $i$ and $\lambda_k$ is the boost for allocation $A_k$. Specifically, each $A_k \in [0, 1]^{n \times m}$ satisfies $\sum_{i=1}^{n}(A_k)_{ij} \leq 1$ for any $j \in [m]$, $\boldsymbol{w} \in \mathbb{R}_+^n$, and $\boldsymbol{\lambda} \in \mathbb{R}^S$. A deterministic AMA refers to the AMA whose parameter $\mathcal{A}$ is fixed by all possible deterministic allocations, and so that $S = (n+1)^m$ (each item can be allocated to any of the $n + 1$ bidders).

Formally, with the parameter set as $(\mathcal{A}, \boldsymbol{w}, \boldsymbol{\lambda})$, denote $\text{asw}(k; V) := \sum_{i=1}^{n} w_i(\boldsymbol{v}_i \cdot (A_k)_i) + \lambda_k$ the affine social welfare for $k$-th allocation under valuation profile $V$ and $\text{asw}_{-i}(k; V) = \text{asw}(k; V) - w_i(\boldsymbol{v}_i \cdot (A_k)_i)$, the allocation and payment rule can be written as

$$g^{\text{AMA}}(V) = A_{k^*} : \quad k^* = \arg\max_{k \in [S]} \text{asw}(k; V),$$

$$p_i^{\text{AMA}}(V) = \frac{1}{w_i}\left(\max_{k \in [S]} \text{asw}_{-i}(k; V) - \text{asw}_{-i}(k^*; V)\right). \tag{AMA}$$

As AMA satisfies DSIC and IR regardless of the chosen parameters (Roberts, 1979; Sandholm & Likhodedov, 2015), the problem of finding the revenue-maximizing AMA with a fixed size of $\mathcal{A}$, $|\mathcal{A}| = S$, can be formulated as an unconstrained optimization.

$$\max_{\mathcal{A}:|\mathcal{A}|=S, \boldsymbol{w}, \boldsymbol{\lambda}} \quad \text{REV}_{\mathcal{F}}^{\text{S-AMA}} := \mathbb{E}_{V \sim \mathcal{F}} \sum_{i=1}^{n} p_i^{\text{AMA}}(V; \mathcal{A}, \boldsymbol{w}, \boldsymbol{\lambda}). \tag{AMA-OPT}$$

Specifically, *we denote $\text{REV}_{\mathcal{F}}^{\text{D-AMA}}$ the optimal revenue when fixing $\mathcal{A}$ to be the set of all deterministic allocations.* To clarify our notation:

- $\text{REV}_{\mathcal{F}}^{\text{S-AMA}}$ refers to the optimal revenue for a **randomized AMA** with menu size pre-set to $S$.
- $\text{REV}_{\mathcal{F}}^{\text{D-AMA}}$ refers to the optimal revenue for a **deterministic AMA** (where $\mathcal{A}$ is fixed to all deterministic allocations).

Recent work on AMA has the advantage of interpretability and strong performance in theory and empirical (Lavi et al., 2003) shows that AMA is "approximately universal" under certain distributions, and recent AMA-based work (Sandholm & Likhodedov, 2015; Curry et al., 2023; Duan et al., 2023; 2026) attain considerable empirical performance when combined with machine learning approaches, even compared with those approximate DSIC auctions.

## 3. Correlation-Aware Affine Maximizer Auctions

This section begins by presenting a bidder-correlated single-item scenario where classic AMAs fail to achieve optimal revenue. We then define Correlation-Aware AMA, a modification that introduces a correlation-aware payment term to enhance AMA's expressiveness while preserving the desirable property of DSIC. The problem of finding the optimal CA-AMA is subsequently formulated as an optimization problem constrained by IR. Finally, we provide a theoretical comparison of the revenue achievable by optimal CA-AMA and classic AMA in single-item auctions.

### 3.1. AMA Fails in Certain Distributions

We begin by analyzing a potential shortcoming of AMA-OPT. In current AMA-based methods (Sandholm & Likhodedov, 2015; Curry et al., 2023; Duan et al., 2023; 2026), AMA parameters $(\mathcal{A}, \boldsymbol{w}, \boldsymbol{\lambda})$ are determined during training and remain fixed at test time to ensure DSIC. Consequently, this static nature prevents the mechanism from further utilizing information from a specific input bidding profile $B$ during evaluation. Specifically, if bidders' valuations are linearly correlated, one bidder's valuation $\boldsymbol{v}_i$ can be inferred from the valuations of others, $V_{-i}$. To illustrate this deficiency, we construct an asymmetric correlated distribution $\mathcal{F}$ where the optimal AMA's revenue can be an arbitrarily small fraction of the optimal revenue.

**Proposition 3.1.** *In single-item auctions, for any number of bidders $n$ and any $\epsilon > 0$, there exists a distribution $\mathcal{F}$ such that $\text{REV}_{\mathcal{F}}^{\text{D-AMA}} \leq \epsilon \cdot \text{REV}_{\mathcal{F}}$. Furthermore, $\text{REV}_{\mathcal{F}}^{\text{S-AMA}} < \text{REV}_{\mathcal{F}}$ for any menu size $S$.*

We refer the reader to Appendix B for the constructed distribution and the complete proof.

### 3.2. Correlation-Aware Payment

Motivated by this failure case of classic AMAs, we propose a modification to address correlated valuation distributions. Specifically, we introduce an additional payment term for each bidder $i$, $p_i^{\text{Cor}}(V_{-i})$, which depends solely on the valuations of other bidders, $V_{-i}$. Formally, the CA-AMA

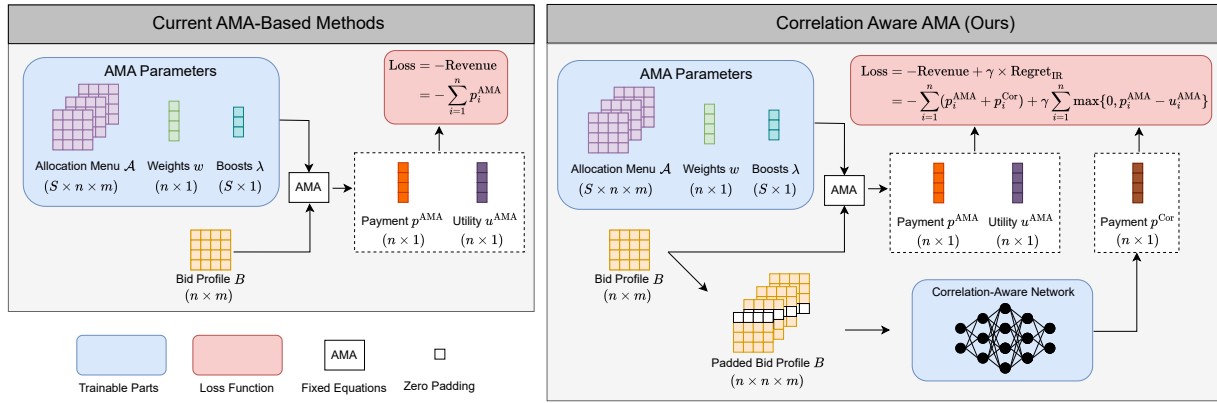

*Figure 1.* The comparison between the optimization for classic Affine Maximizer Auctions (AMAs) and our proposed Correlation Aware AMA (CA-AMA). In classic AMA-based methods (Sandholm & Likhodedov, 2015; Curry et al., 2023; Duan et al., 2023; 2026), we only optimize the AMA parameters to improve the revenue. To enhance AMA's performance under bidder-correlated distributions, we introduce a correlation-aware payment $p^{\text{Cor}}$ and hence add a $\text{Regret}_{\text{IR}}$ term in our loss function.

mechanism is defined as:

$$g^{\text{CA}}(V; \mathcal{A}, \boldsymbol{w}, \boldsymbol{\lambda}) = g^{\text{AMA}}(V; \mathcal{A}, \boldsymbol{w}, \boldsymbol{\lambda}),$$

$$p_i^{\text{CA}}(V; \mathcal{A}, \boldsymbol{w}, \boldsymbol{\lambda}, p^{\text{Cor}}) = p_i^{\text{AMA}}(V; \mathcal{A}, \boldsymbol{w}, \boldsymbol{\lambda}) + p_i^{\text{Cor}}(V_{-i}).$$

For any bidder $i$, since $p_i^{\text{Cor}}(V_{-i})$ depends only on other bidders' valuations, it acts as a constant from bidder $i$'s perspective when determining their optimal bid. Thus, the optimal bidding strategy remains unchanged from that in a classic AMA. Therefore, CA-AMA inherits the DSIC property from AMA.

**Proposition 3.2.** *For any $\mathcal{A}$, $\boldsymbol{w}$, $\boldsymbol{\lambda}$ and correlation-aware function $p^{Cor}$, the CA-AMA mechanism $(g^{CA}, p^{CA})$ satisfies DSIC.*

However, IR can be violated if $p_i^{\text{Cor}}(V_{-i})$ is set inappropriately high. Considering this, we formulate the problem of finding the optimal CA-AMA as an IR-constrained optimization problem:

$$\max_{\mathcal{A}:|\mathcal{A}|=S, \boldsymbol{w}, \boldsymbol{\lambda}, p^{\text{Cor}}} \text{REV}_{\mathcal{F}}^{\text{S-CA}} := \mathbb{E}_{V \sim \mathcal{F}} \sum_{i=1}^{n} p_i^{\text{CA}}(V)$$

$$\text{s.t.} \quad \text{The mechanism } (g^{\text{CA}}, p^{\text{CA}}) \text{ satisfies IR.}$$
$$\text{(CA-AMA-OPT)}$$

Note that we omit the parameters of CA-AMA for the above $p^{\text{CA}}(\cdot)$ function. Similar to the notation for AMA, we denote $\text{REV}_{\mathcal{F}}^{\text{D-CA}}$ to be the optimal revenue obtained by CA-AMA with $\mathcal{A}$ fixed to be the set of all deterministic allocations. To highlight the importance of this formulation for correlated distributions, we analyze the relationship between the optimal revenues from AMA and CA-AMA. Our analysis primarily focuses on single-item auctions; we will also discuss the challenges in extending these theoretical guarantees to multi-item settings. The empirical performance of CA-AMA in multi-item auctions is demonstrated in Section 5.

Clearly, for any distribution $\mathcal{F}$, $\text{REV}_{\mathcal{F}}^{\text{CA}} \geq \text{REV}_{\mathcal{F}}^{\text{AMA}}$, since setting $p_i^{\text{Cor}}(V_{-i}) = 0$ for all $i$ allows CA-AMA to replicate any classic AMA. We then present cases where this relationship can be further characterized.

**Theorem 3.3.** *In single-item auctions, for any number of bidders $n$:*

- *If $\mathcal{F}$ is bidder-independent, then $\text{REV}_{\mathcal{F}}^{D\text{-}CA} = \text{REV}_{\mathcal{F}}^{D\text{-}AMA}$.*
- *For any $\epsilon > 0$, there exists a distribution $\mathcal{F}$ such that $\text{REV}_{\mathcal{F}}^{D\text{-}AMA} \leq \epsilon \cdot \text{REV}_{\mathcal{F}}$, while $\text{REV}_{\mathcal{F}}^{D\text{-}CA} = \text{REV}_{\mathcal{F}}$. Furthermore, $\text{REV}_{\mathcal{F}}^{S\text{-}AMA} < \text{REV}_{\mathcal{F}}$ for any menu size $S$.*

This result indicates that introducing the $p_i^{\text{Cor}}(V_{-i})$ term offers no benefit over classic AMAs in bidder-independent single-item auctions when considering deterministic mechanisms. The second part of the theorem utilizes the same constructed distribution as in Proposition 3.1. Under such correlated distributions, CA-AMA demonstrates significantly greater expressiveness than classic AMAs, achieving optimal revenue where AMAs fail.

**Extension to Multi-Item Auctions.** While Theorem 3.3 pertains to single-item auctions, we believe similar results hold for multi-item auctions, particularly under **negative correlation** between bidders' valuations. The intuition follows from our earlier observation: in classic AMA, a bidder's payment can only be a non-decreasing function of other bidders' valuations. Since we expect a bidder's payment to increase with its own valuation, classic AMA cannot express this property under negative correlation, leading to revenue loss. The distributions used in our experiments (Dirichlet Value Share and Linear Mixture) are not merely corner cases but have representative significance for negatively correlated settings.

Theoretically, the constructive proof of the second part of Theorem 3.3 directly extends to multi-item settings: if bidders have *identical* valuations across all items following our constructed single-item distribution, deterministic AMA performs arbitrarily poorly and randomized AMA remains suboptimal, whereas CA-AMA achieves optimality. For independent valuations across items, proving "CA-AMA = AMA" remains a conjectured future direction. As we have spent considerable time attempting this proof, we discuss several theoretical challenges around Line 190: the main difficulty is that characterizing the optimal parameters even for classic AMA is hard, making the comparison truly challenging. Therefore, we primarily validate the performance of CA-AMA in multi-item settings empirically in Section 5.

So far, we have introduced the CA-AMA framework, formulated its optimization problem, and theoretically analyzed its potential for revenue improvement over classic AMAs. The subsequent section will propose a data-driven algorithm for optimizing CA-AMA.

## 4. Optimization of CA-AMA

This section details the procedure for optimizing the CA-AMA. Firstly, we design a loss function for Problem CA-AMA-OPT within a data-driven framework. We then propose a two-stage training algorithm to optimize both the AMA parameters and the correlation-aware payments $p^{\text{Cor}}$, which is summarized in Algorithm 1. We also introduce a post-processing method to let CA-AMA achieve strict ex-post IR. Finally, we provide theoretical support for our method by establishing the continuity of the optimal $p^{\text{Cor}}$ under mild assumptions and proving that the generalization error of the IR violation is bounded.

### 4.1. Loss Function and Training

To address Problem CA-AMA-OPT, we parameterize the AMA components $(\mathcal{A}, \boldsymbol{w}, \boldsymbol{\lambda})$ and the correlation-aware payments $p^{\text{Cor}}$ using neural networks with parameters $\theta$ and $\phi$, respectively. Optimization is performed via the minimization of a loss function that incorporates a trade-off between revenue maximization and IR violations.

We define the $\text{Regret}_{\text{IR}}$ for a single data point $V$ as

$$\text{Regret}_{\text{IR}}(V) := \sum_{i=1}^{n} \max\{0, -u_i^{\text{CA}}(v_i, V)\},$$

where $u_i^{\text{CA}}(V) = u_i^{\text{AMA}}(V) - p_i^{\text{Cor}}(V_{-i})$. Given a dataset $D = \{V^{(k)}\}_{k=1}^{K}$, the empirical loss for the dataset is:

$$\mathcal{L}(\theta, \phi) := \sum_{k=1}^{K} \left( -\text{Revenue}(V^{(k)}) + \gamma \cdot \text{Regret}_{\text{IR}}(V^{(k)}) \right). \tag{1}$$

The revenue term comprises payments from both the AMA and the core price components, defined as: $\text{Revenue}(V) = \sum_{i=1}^{n} (p_i^{\text{AMA}}(V) + p_i^{\text{Cor}}(V_{-i}))$. $\gamma$ is the hyperparameter that adjusts the strength of the IR penalty. Following Ivanov et al. (2022), $\gamma$ is updated iteratively based on a target regret $R_{\text{target}}$:

$$\gamma_{t+1} = \text{clip}\left(\gamma_t + \gamma_\Delta \left(\log R(D) - \log R_{\text{target}}\right), 1, \bar{\gamma}\right). \tag{2}$$

**Comparison with Other Regret Terms.** Introducing a regret term into the loss function is a common technique for handling constraints in auction optimization (Dütting et al., 2023; Wang et al., 2024), as it transforms a hard constraint into a soft penalty that is more amenable to gradient-based methods. We emphasize that IR-regret and IC-regret serve fundamentally different purposes: IC-regret enforces incentive compatibility constraints, whereas IR-regret only controls individual rationality violations. Since CA-AMA guarantees DSIC by construction, it does not need to compute IC-regret during training, which yields significant *computational efficiency* advantages.

Specifically, computing IC-regret as used in RegretNet (Dütting et al., 2023) necessitates running many additional auctions to find the approximate best bid value. Similarly, GemNet (Wang et al., 2024) involves complex integer programming to ensure strict feasibility. In contrast, IR-regret can be estimated alongside the revenue in a single forward pass of the mechanism, making it computationally efficient. Furthermore, IR-regret is more flexible to be adjusted: we can easily reduce $p_i^{\text{Cor}}$ by a constant to lower the IR-regret after training, or even enforce strict IR using the post-processing transformation proposed in the following. Adjusting the degree of IC-regret or feasibility regret post-training is non-trivial and typically requires retraining.

**Two-Stage Training.** To attain a mechanism with high revenue and low $\text{Regret}_{\text{IR}}$, we propose a two-stage optimization procedure: mutual training followed by post-training.

In the **mutual training** stage, the parameters $\theta$ (for AMA components) and $\phi$ (for $p^{\text{Cor}}$) are jointly trained. As the revenue is non-differentiable to the AMA parameters, we follow the previous methods (Curry et al., 2023; Duan et al., 2023) to replace the argmax in the allocation rule of AMA with a softmax approximation. The primary objective of mutual training is to find AMA parameters that are close to optimal for the combined objective. However, as the true AMA utility and the actual regret of IR violation are just estimated by softmax, they may not precisely meet the target $R_{\text{target}}$ after this stage.

Therefore, a subsequent **post-training** stage is introduced to further refine $p^{\text{Cor}}$. In this stage, the AMA parameters are frozen. Since gradients of $\theta$ are not required, the exact

AMA payments $p_i^{\text{AMA}}$ and utilities $u_i^{\text{AMA}}$ are used in the loss calculation for this stage. The rationale for fixing $\theta$ is that mutual training is assumed to have found a near-optimal configuration for the core AMA structure; post-training then performs a more precise adjustment of $p_i^{\text{Cor}}$.

Furthermore, for applications requiring **strict ex-post IR**, our trained CA-AMA can be easily adapted. We introduce a simple **post-processing step**: after the mechanism determines outcomes, any bidder facing a negative utility can choose to opt out, receiving zero allocation and making a zero payment. This ensures all participating bidders have non-negative utility, thus making the mechanism strictly IR. Crucially, this transformation preserves the DSIC property. A bidder's utility becomes $\max\{u_i, 0\}$. Since $\max\{\cdot, 0\}$ is a non-decreasing function, any bid that maximizes the original utility $u_i$ also maximizes the post-transformation utility. Therefore, truthful bidding remains a dominant strategy.

### 4.2. Theoretical Characterizations

To conclude this section, we present theoretical results that support the validity and tractability of our optimization approach. Our theoretical analysis focuses on the novel aspects compared to classic AMA: the correlation-aware term $p^{\text{Cor}}$ and the Regret$_{\text{IR}}$ component of the loss.

**Continuity of Optimal $p^{\text{Cor}}$.** For bidder $i$'s correlation-aware payment $p_i^{\text{Cor}}$, to maximize revenue subject to IR ($u_i^{\text{CA}} \geq 0$, which implies $u_i^{\text{AMA}}(V) - p_i^{\text{Cor}}(V_{-i}) \geq 0$), the largest such $p_i^{\text{Cor}}(V_{-i})$ is given by:

$$p_i^{\text{OPT-core}}(V_{-i}) := \inf_{\boldsymbol{v}_i \in \text{supp}(\mathcal{F}_i(V_{-i}))} u_i^{\text{AMA}}((\boldsymbol{v}_i, V_{-i}); \mathcal{A}, \boldsymbol{w}, \boldsymbol{\lambda}).$$

This means that to maximize revenue subject to IR, $p_i^{\text{Cor}}(V_{-i})$ should ideally be set to the minimum utility bidder $i$ would receive from the AMA mechanism. Intuitively, if $p_i^{\text{Cor}}(V_{-i})$ exceeds this value, IR is violated; if it is less, the revenue is sub-optimal. We then establish continuity properties for this $p_i^{\text{OPT-core}}$.

**Theorem 4.1.** *The target function $p_i^{OPT\text{-}core}$ is continuous with respect to the AMA parameters $\mathcal{A}$, $\boldsymbol{w}$, and $\boldsymbol{\lambda}$. Furthermore, assume that there exists a constant $C_H > 0$ such that for all $V_{-i}, V'_{-i}$, the Hausdorff distance $h(supp(\mathcal{F}_i(V_{-i})), supp(\mathcal{F}_i(V'_{-i}))) \leq C_H \|V_{-i} - V'_{-i}\|$, then $p_i^{OPT\text{-}core}$ is also continuous with respect to $V_{-i}$.*

This result demonstrates that the optimal $p_i^{\text{Cor}}(V_{-i})$ is continuous with respect to both the AMA parameters and the input $V_{-i}$ under these mild assumptions. This continuity supports the feasibility of parameterizing $p_i^{\text{Cor}}$ with a neural network, which is a universal approximator for any continuous function (Hornik et al., 1989; Cybenko, 1989).

*Remark* 4.2 (Structural Characterization of Correlation-Aware Payment). The target correlation-aware payment has an explicit structural form. For a fixed menu $\mathcal{A}$

of size $K$, the bidder's utility under AMA is determined by the difference between two terms: the maximum affine social welfare over all possible allocations, and the maximum affine social welfare when the current bidder is excluded. Each term is the maximum of linear functions of bidder valuations. Thus, the optimal correlation-aware payment, which equals the minimum possible utility, is **a difference of two maxima of linear functions**. This structure characterizes the theoretical target function; in our experiments, we use an unconstrained neural network for $p^{\text{Cor}}$ as it is easy to implement and demonstrates strong empirical performance.

**Generalization Bound of Regret$_{\text{IR}}$.** We next provide a guarantee on the generalization of the IR regret term. This addresses the concern of whether a mechanism trained on a finite dataset will exhibit similarly low regret on unseen data drawn from the true underlying distribution $\mathcal{F}$. Specifically, we aim to show that the empirical Regret$_{\text{IR}}$, computed on the training set, is a reliable proxy for the true expected Regret$_{\text{IR}}$ under $\mathcal{F}$. Our analysis considers the post-training stage, where AMA parameters are fixed, and only $p^{\text{Cor}}$ is being learned. The following theorem bounds the difference between the empirical and expected Regret$_{\text{IR}}$.

**Theorem 4.3** (Informal version of Theorem C.1)**.** *For each $i \in [n]$, let $p_i^{Cor}$ be the output of a 3-layer ReLU network whose weights have bounded spectral norms. Then, for any AMA parameters $(\mathcal{A}, \boldsymbol{w}, \boldsymbol{\lambda})$, distribution $\mathcal{F}$ and i.i.d. sample $D = \{V^{(1)}, \dots, V^{(K)}\} \sim \mathcal{F}^K$, the following inequality holds uniformly over all such networks (i.e., all choices of parameters $\theta$) with probability $1 - \delta$:*

$$\left| \frac{1}{K} \sum_{k=1}^{K} Regret_{IR}(V^{(k)}) - \mathbb{E}_V[Regret_{IR}(V)] \right|$$
$$\leq O\left( \sqrt{\frac{\log(1/\delta)}{K}} \right).$$

This result guarantees that minimizing the empirical regret on a sufficiently large training set allows us to control the true expected regret of the learned mechanism. Combined with the continuity of $p^{\text{OPT-core}}$, these results provide theoretical grounding for our proposed training algorithm. In the next section, we will evaluate the CA-AMA framework and training algorithm empirically.

## 5. Experimental Results

This section presents the implementation and empirical evaluation of CA-AMA[1]. We first compare revenue against baselines across multiple distributions and auction settings. We then visualize a perfectly correlated case to highlight the limitations of classic AMA and the advantage of CA-AMA.

---

[1]The implementation is available at https://github.com/Haoran0301/CA-AMA.git

*Table 1.* Revenue performance of CA-AMA and baseline methods under various bidder valuation distributions. CA-AMA consistently outperforms other methods in most scenarios. All results are averaged over 5 different random seeds. The number in parentheses indicates the average $\text{Regret}_{\text{IR}}$ of CA-AMA under the IR constraint. The column "Neg. Utility" shows the fraction of samples with negative utility before post-processing.

| Setting | Item-CAN | VCG | Randomized AMA | CA-AMA ($\text{Regret}_{\text{IR}}$) | CA-AMA (Ex-post IR) | Neg. Utility |
|---|---|---|---|---|---|---|
| *Dirichlet Value Share ($\alpha = 0.5$)* | | | | | | |
| $2 \times 2$ | 0.7874 | 0.2702 | 0.7480 | 0.8532 (0.0011) | **0.8261** | 0.0200 |
| $2 \times 5$ | 1.9685 | 0.6774 | 1.8808 | 2.3663 (0.0048) | **2.2694** | 0.0393 |
| $3 \times 10$ | 3.3121 | 1.7572 | 3.1363 | 3.6205 (0.0031) | **3.5623** | 0.0120 |
| $5 \times 5$ | 1.2674 | 0.9413 | 1.2123 | 1.3209 (0.0009) | **1.3084** | 0.0047 |
| *Dirichlet Value Share ($\alpha = 2.0$)* | | | | | | |
| $2 \times 2$ | 0.6690 | 0.4676 | 0.6516 | 0.7131 (0.0006) | **0.6995** | 0.0129 |
| $2 \times 5$ | 1.6725 | 1.1723 | 1.7362 | 1.9437 (0.0028) | **1.8934** | 0.0239 |
| $3 \times 10$ | 2.6461 | 2.2796 | 2.8706 | 2.9448 (0.0009) | **2.9292** | 0.0046 |
| $5 \times 5$ | 0.9820 | 0.9384 | 1.0239 | 1.0406 (0.0005) | **1.0330** | 0.0081 |
| *Linear Mixture ($\alpha = 0.6$)* | | | | | | |
| $2 \times 5$ (Sym) | 2.2955 | 1.6963 | 2.2923 | 2.6305 (0.0036) | **2.5628** | 0.0182 |
| $2 \times 5$ (Asym) | 1.5715 | 0.6011 | 1.7135 | 1.9359 (0.0052) | **1.8553** | 0.0267 |
| *Linear Mixture ($\alpha = 0.8$)* | | | | | | |
| $2 \times 5$ (Sym) | 2.7655 | 1.4914 | 2.4239 | 3.0837 (0.0034) | **2.9643** | 0.0156 |
| $2 \times 5$ (Asym) | 1.8390 | 0.5767 | 1.6677 | 2.2028 (0.0028) | **2.1124** | 0.0132 |

These experimental results substantiate the theoretical results and demonstrate the practical advantages of CA-AMA. All experiments are conducted on a single NVIDIA A800 GPU with 80GB of memory.

Our primary comparison is between CA-AMA and **Randomized AMA**. Results for Randomized AMA are obtained using the state-of-the-art neural optimization framework AMenuNet (Duan et al., 2023). We also adapt Conditional Auction Net (CAN) (Huo et al., 2025) to the multi-item setting by applying it independently to each item; we refer to this extension as **Item-CAN**. The classic **VCG** auction (Vickrey, 1961) is included as an additional baseline. Although GemNet (Wang et al., 2024) also provides strict DSIC guarantees, we exclude it due to implementation complexity in multi-bidder scenarios.

For implementation, we follow the over-parameterization strategy for AMA parameters used in AMenuNet (Duan et al., 2023). We refer the reader to Duan et al. (2023) for details of network architectures as this is not a primary focus of our work. The correlation-aware payment component $p^{\text{Cor}}$ is implemented as a three-layer ReLU MLP. The menu size $|\mathcal{A}|$ is the same between Randomized AMA and CA-AMA within each auction configuration and scaled with problem size. The IR regret target is $R_{\text{target}} = 0.001$; the penalty coefficient is initialized with $\gamma_0 \in \{3, 5, 10\}$, updated using learning rate $\gamma_\Delta = 0.01$, and capped at $\bar{\gamma} = 20$. The softmax temperature during mutual training is 500. We run 32,000 iterations and the batch size is 2,048 for smaller settings and 1,024 for larger settings. For CA-AMA, we balance the training iterations between mutual training and

post-training within the total iterations. A fixed test set of 20,000 samples is used for evaluation.

### 5.1. Multi-Item Bidder-Correlated Auctions

In this subsection, we evaluate our mechanism across two distinct, representative valuation distributions designed to model bidder correlations. These distributions are the **Dirichlet Value Share** model, which generates complex support-level correlations, and the **Linear Correlation Mixture** model, which introduces a more direct but probabilistic linear dependency.

**Dirichlet Value Share.** This model generates negatively correlated valuations by assuming bidders draw their values from a shared, latent total value for each item. The generation process for each item $j$ is as follows: (1) A latent total value $T_j$ is drawn from a uniform distribution, $T_j \sim U[0.5, 1]$. (2) A share vector for $n$ bidders, $w_j = (w_{1j}, \ldots, w_{nj})$, is drawn from a symmetric Dirichlet distribution, $w_j \sim \text{Dirichlet}(\alpha, \ldots, \alpha)$, where $\sum_{i=1}^{n} w_{ij} = 1$. (3) The final valuation for bidder $i$ is calculated as $v_{ij} = w_{ij} \cdot T_j$. The parameter $\alpha$ controls the correlation strength: a small $\alpha$ leads to a sparse share allocation and strong negative correlation, while a larger $\alpha$ results in more uniform shares and weaker correlation.

**Linear Correlation Mixture.** This model investigates scenarios with a more explicit, probabilistic linear correlation between two bidders. For each item $j$, the first bidder's valuation, $v_{1j}$, is sampled from $U[0, 1]$. The second bidder's valuation, $v_{2j}$, with probability $\alpha$, is linearly dependent on

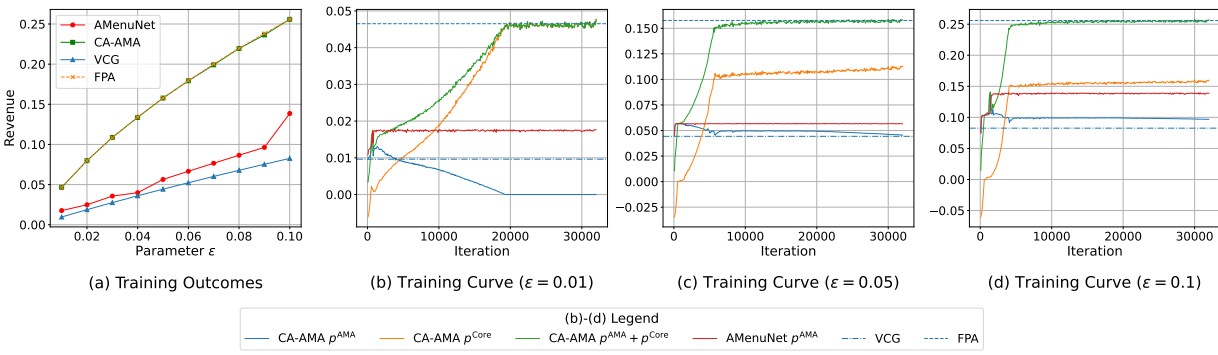

*Figure 2.* The revenue results and training curves of CA-AMA and Randomized AMA (implemented by AMenuNet (Duan et al., 2023)) in auctions with the first bidder's valuation $v_1$ following equal revenue distribution on $[\epsilon, 1]$ and the second bidder's valuation $v_2 = \frac{\epsilon}{1-\epsilon}(1-v_1)$. As the final Regret$_{\text{IR}}$ in all cases is less than $1e-5$, it is not plotted in the figure.

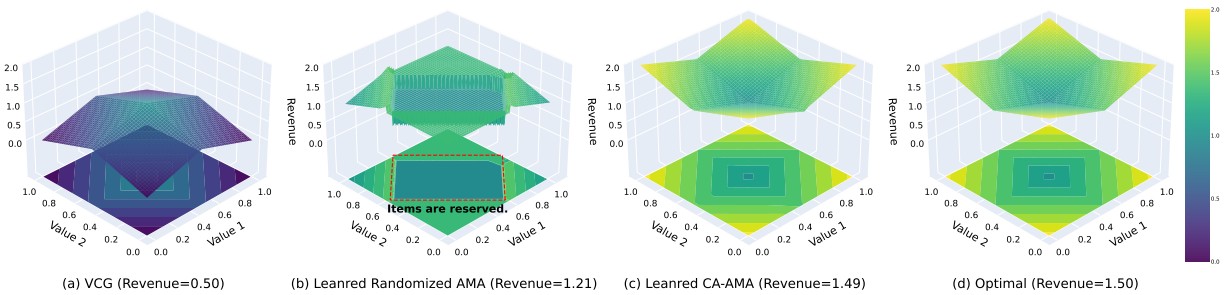

*Figure 3.* Revenue surfaces of learned CA-AMA and Randomized AMA in a 2-bidder, 2-item perfectly negative linear scenario ($v_{21} = 1 - v_{11}$ and $v_{22} = 1 - v_{12}$). Bidder 1's valuations ($v_{11}, v_{12}$) are on the x-y axes; revenue is on the z-axis. CA-AMA closely approximates the optimal revenue surface, while Randomized AMA often reserves items and has sub-optimal revenue.

$v_{1j}$; otherwise, with probability $1 - \alpha$, it is drawn independently. We consider two cases. In the **Symmetric Case**, with probability $\alpha$, $v_{2j} = 1 - v_{1j}$; otherwise, $v_{2j}$ is drawn independently from $U[0, 1]$. In the **Asymmetric Case**, with probability $\alpha$, $v_{2j} = (1 - v_{1j})/4$; otherwise, $v_{2j}$ is drawn independently from $U[0, 1/4]$. In our experiments, we test correlation probabilities of $\alpha \in \{0.6, 0.8\}$.

Table 1 summarizes the revenue results across all auction configurations. For CA-AMA, we report revenue both before and after the post-processing step that ensures strict ex-post IR. The results show that CA-AMA consistently outperforms all baselines in revenue, even under the strict ex-post IR constraint, demonstrating its robustness across diverse auction settings. Furthermore, before post-processing, CA-AMA's IR regret is consistently near $0.001$, confirming our training algorithm's effectiveness in satisfying IR constraints. Revenue improvements are most pronounced in scenarios with stronger correlations, such as smaller $\alpha$ in the Dirichlet model and larger $\alpha$ in the Linear Correlation Mixture model. Among the baselines, Item-CAN performs better than Randomized AMA under stronger correlations but underperforms in weaker settings, consistent with its design to utilize correlation information. However, CA-AMA

shows significantly greater improvement than both, underscoring its superior ability to leverage bidder correlations for revenue enhancement.

Beyond revenue, we note that the correlation-aware payment component does not significantly increase computational complexity. Table 2 compares the training times of CA-AMA and Randomized AMA, showing that they are comparable despite the added complexity.

### 5.2. Perfectly Correlated Valuations

This section examines a two-bidder auction with perfectly linear correlations. While this is an extreme case, it serves to clearly illustrate the limitations of the classic AMA framework compared to CA-AMA.

**Equal Revenue Distribution.** We first consider a single-item auction where the first bidder's valuation $v_1$ follows an equal revenue distribution on $[\epsilon, 1]$, and the second bidder's valuation is perfectly correlated as $v_2 = \frac{\epsilon}{1-\epsilon}(1-v_1)$. This setup mirrors the construction in the proof of Theorem 3.3, where we show that for sufficiently small $\epsilon$, the gap between optimal deterministic AMA and CA-AMA can be arbitrarily large. We explore the performance of CA-AMA and Ran-

*Table 2.* Training time for 32,000 iterations of Randomized AMA and CA-AMA across settings , measured on a single A800 GPU.

| Setting | $2 \times 2$ | $2 \times 5$ | $3 \times 10$ | $5 \times 5$ |
|---|---|---|---|---|
| Randomized AMA | 14 min | 26 min | 52 min | 1 h 50 min |
| CA-AMA | 15 min | 27 min | 54 min | 1 h 55 min |

domized AMA under varying values of $\epsilon$. Figure 2 presents the training dynamics of CA-AMA and Randomized AMA (implemented via AMenuNet (Duan et al., 2023)). For comparison, we also plot the revenue of VCG and the first-price auction (FPA), the latter extracting full surplus and thus representing the optimal revenue alongside the theoretical optimum for Randomized AMA. The results demonstrate that CA-AMA successfully converges to the optimal revenue, significantly outperforming Randomized AMA. By examining the payment components $p^{\mathrm{Cor}}$ and $p^{\mathrm{AMA}}$, we observe that CA-AMA effectively identifies the correlation structure in the distribution, with $p^{\mathrm{Cor}}$ dominating the total payment in all cases. Although the AMA-derived revenue (CA-AMA $p^{\mathrm{AMA}}$) is lower than that of AMenuNet, the total revenue from CA-AMA ($p^{\mathrm{AMA}} + p^{\mathrm{Cor}}$) is substantially higher.

**Perfect Negative Linear Correlation.** In Figure 3, we visualize the revenue surfaces of the learned CA-AMA and Randomized AMA mechanisms in a 2-bidder, 2-item setting with perfect negative linear correlation: $v_{21} = 1 - v_{11}$ and $v_{22} = 1 - v_{12}$. The figure displays the extracted revenue (z-axis) as a function of bidder 1's valuations for the two items ($v_{11}$ on the x-axis, $v_{12}$ on the y-axis). CA-AMA's learned revenue surface closely approximates the optimal outcome, highlighting its ability to learn near-optimal allocation and payment rules. In contrast, while Randomized AMA improves upon VCG, it deviates significantly from the optimal surface. Notably, it frequently reserves items even in high-valuation regions, underscoring its inherent limitations in correlated environments.

## 6. Conclusion

In this paper, we address the critical limitation of existing AMAs in bidder-correlated settings, where their inherent VCG-style payment rules restrict flexibility and lead to suboptimal revenue extraction. To overcome this, we introduce the CA-AMA, an extended mechanism incorporating an additional correlation-aware payment term. We demonstrate that CA-AMA inherently preserves the DSIC property and can theoretically achieve optimal revenue in single-item auctions under certain correlated distributions where classic AMAs perform arbitrarily poorly. Furthermore, we propose a two-stage training algorithm for optimizing CA-AMA, supported by theoretical guarantees on continuity and generalization. Our extensive experimental evaluations across diverse valuation distributions confirm the empirical

effectiveness of CA-AMA, showcasing its ability to achieve significantly improved revenue compared to AMAs with comparable computational efficiency.

## Acknowledgement

This work was supported by the Natural Science Foundation of China (Grant No. 62572010).

## Impact Statement

This paper presents work whose goal is to advance the field of algorithmic game theory. While CA-AMA advances automated mechanism design theoretically, its practical deployment in environments like ad auctions carries significant economic implications.

**Broader Impact and Ethical Considerations.** By leveraging peer information to set personalized, correlation-aware payments ($p^{\mathrm{core}}$), the mechanism inherently engages in *algorithmic price discrimination*. Bidders with identical true valuations may face different effective reserve prices depending on the inferred behavior of others. Furthermore, while CA-AMA maintains DSIC for non-cooperative individuals, it may inadvertently encourage unwanted strategic behavior in highly correlated environments. Bidders aware of these dynamics might be incentivized to form collusive rings or employ artificial bid-shading to manipulate payment terms, highlighting a crucial trade-off between revenue maximization and market fairness.

Mechanism designers deploying CA-AMA should carefully consider these implications, particularly in applications where fairness and transparency are paramount. We encourage future research to explore regulatory frameworks and algorithmic modifications that can mitigate these concerns while preserving the revenue benefits of correlation-aware mechanisms.

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

# A. Detailed Related Work

## A.1. Affine Maximizer Auctions

Affine maximizer auctions generalize the seminal VCG auction by assigning positive weights to bidders and boost variables to allocations, modifying the objective to maximize affine social welfare. By adjusting these parameters, AMAs can represent a wide range of auction mechanisms while inherently satisfying dominant strategy incentive compatibility and individual rationality. There are previous studies consider different subclasses of AMA for more desirable theoretical characterizations. These subclasses includes Virtual Valuations Combinatorial Auctions (VVCAs) (Likhodedov & Sandholm, 2004; Likhodedov et al., 2005; Sandholm & Likhodedov, 2015), $\lambda$-auctions (Jehiel et al., 2007), mixed bundling auctions (Tang & Sandholm, 2012), and bundling-boosted auctions (Balcan et al., 2021).

The expressiveness of AMAs in comparison to arbitrary auction mechanisms has been formally analyzed in (Lavi et al., 2003), which showed that AMAs can represent optimal auctions under certain conditions. Beyond expressiveness, there are studies consider the optimization of an AMA: Sandholm & Likhodedov (2015) presented optimization methods for finding optimal AMA mechanisms, while Balcan et al. (2016; 2018) studied the sample complexity required to learn such mechanisms. More recently, differentiable optimization techniques have been applied to this setting. For example, LotteryAMA (Curry et al., 2023) and AMenuNet (Duan et al., 2023) introduce differentiable approaches to optimize AMA-based auctions using neural networks.

Our work proposes a new framework, CA-AMA, that extends the classical AMA by incorporating bidder correlations. We theoretically characterize its expressiveness relative to traditional AMA in single-item settings and empirically evaluate optimization algorithms for learning revenue-optimal CA-AMA mechanisms across various distributional settings.

## A.2. Differentiable Economics for Auctions

Differentiable economics, which leverages neural networks as flexible function approximators and optimizes them using gradient-based methods, is a recent and active line of research in automated mechanism design Existing work in this area for revenue maximization can be broadly categorized into *characterization-free* and *characterization-based approaches*.

Characterization-free methods do not assume a predefined structure for the mechanism. The foundational work, Regret-Net (Dütting et al., 2023), implements the allocation and payment rules as neural networks conditioned on bid profiles. Its loss function jointly optimizes revenue and penalizes violations of DSIC and IR. Building on this, Feng et al. (2018) incorporated budget constraints, while Golowich et al. (2018) generalized the framework to handle various objectives and constraints. Rahme et al. (2021b) reframed the design problem as a two-player game with a more efficient loss. Further extensions include PreferenceNet (Peri et al., 2021), which incorporates fairness preferences, and EquivariantNet (Rahme et al., 2021a), a permutation-equivariant architecture tailored for symmetric auctions. Transformer-based methods, such as those introduced by Ivanov et al. (2022) and Duan et al. (2022), improve performance in settings with contextual information. Hertrich et al. (2023) applied mode connectivity to provide a theoretical explanation for the empirical success of differentiable economics. The combinatorial auction extensions CANet and CAFormer (Pham et al., 2025) bring these ideas into richer valuation domains.

Characterization-based approaches, by contrast, restrict optimization to a predefined family of mechanisms. AMAs are particularly suitable for this due to their inherent satisfaction of DSIC and IR. LotteryAMA (Curry et al., 2023) introduces randomized allocation menus over AMA structures, which simplifies optimization. AMenuNet (Duan et al., 2023) builds upon this with a more expressive architecture and applies it to contextual auctions. Further developments include contextual AMAs for ad auctions (Li et al., 2024), dynamic AMA designs (Curry et al., 2024), and zeroth-order optimization for deterministic AMA mechanisms (Duan et al., 2026). In addition, menu-based mechanisms have also been treated with differentiable tools. MenuNet (Shen et al., 2019) optimizes revenue for single-bidder auctions, while GemNet (Wang et al., 2024) extends to multi-bidder cases by incorporating over-allocation penalties and post-processing using mixed-integer linear programming.

Our work fits within the characterization-based paradigm. We extend AMA to define CA-AMA, a mechanism that incorporates bidder correlations through a novel correlation-aware payment rule. This new structure retains the theoretical guarantees of classic AMA while significantly improving revenue, both in theory and in practice.

### A.3. Auctions with Bidder Correlations

Modeling bidder correlation is a critical aspect of realistic auction settings. The foundational Crémer-McLean results (Crémer & McLean, 1985; 1988) demonstrate that under certain distributional conditions, it is possible to design mechanisms that are DSIC, interim IR, and extract the full surplus. However, like the Myerson auction, these mechanisms assume full knowledge of the valuation distribution and thus are primarily theoretical.

Subsequent work has relaxed this assumption by exploring scenarios in which the auctioneer has incomplete information. Fu et al. (2014), Albert et al. (2017), and Yang & Bei (2021) studied the sample complexity needed to approximate Crémer-McLean-style mechanisms from empirical data. Because computing the optimal mechanism under general correlated settings is NP-hard, approximation algorithms have also been proposed. For instance, Dobzinski et al. (2011) designed polynomial-time mechanisms that achieve provable approximation guarantees under correlated priors. In contrast, Papadimitriou & Pierrakos (2015) and Caragiannis et al. (2016) provided upper bounds by constructing distributions where any polynomial-time algorithm performs poorly. There are works analyzing the mechanism's robustness to correlation. Bei et al. (2019) studied the correlation-robust design problem, while Zhang (2021) and He & Li (2022) showed that the second-price auction is asymptotically optimal in worst-case correlated environments.

More recently, Prasad et al. (2023) modified VCG auction with side information, achieving improvements in both revenue and social welfare. This is followed by Prasad et al. (2025); Balcan et al. (2025), who studied further game theoretical properties of such mechanisms. The idea of leveraging side information is highly related to our CA-AMA framework, which also utilizes additional information from bidder correlations to enhance revenue.

These studies predominantly focus on theoretical designs for single-item auctions. In contrast, our goal is to demonstrate both the theoretical and empirical benefits of CA-AMA in richer combinatorial settings. The most closely related works are Huo et al. (2025) and Feldman & Lavi (2021). The former proposes a score-based payment rule, optimized through a max-min neural architecture to approximate optimal revenue in single-item settings. The latter provides a theoretical analysis of the gap between ex-post and ex-interim IR mechanisms, showing that AMA can perform arbitrarily poorly in the presence of correlations. Our results extend this by showing that the performance gap holds even when comparing to ex-post IR mechanisms, and we demonstrate that CA-AMA overcomes this gap.

## B. Omitted Proofs in Section 3

**Proposition 3.1.** *In single-item auctions, for any number of bidders $n$ and any $\epsilon > 0$, there exists a distribution $\mathcal{F}$ such that $REV_{\mathcal{F}}^{D\text{-}AMA} \leq \epsilon \cdot REV_{\mathcal{F}}$. Furthermore, $REV_{\mathcal{F}}^{S\text{-}AMA} < REV_{\mathcal{F}}$ for any menu size $S$.*

*Proof.* See the proof of Theorem 3.3. $\qquad\square$

**Proposition 3.2.** *For any $\mathcal{A}$, $\boldsymbol{w}$, $\boldsymbol{\lambda}$ and correlation-aware function $p^{Cor}$, the CA-AMA mechanism $(g^{CA}, p^{CA})$ satisfies DSIC.*

*Proof.* We verify the DSIC property by definition. For any bidder $i$, its true valuation $\boldsymbol{v}_i$, other bidders' bid $V_{-i}$ and possible bid $\boldsymbol{b}_i$, we define $A_{k^*} := g^{\mathrm{AMA}}(\boldsymbol{v}_i, V_{-i})$, $A_{k'^*} := g^{\mathrm{AMA}}(\boldsymbol{b}_i, V_{-i})$ and $A_{k^*} := g^{\mathrm{AMA}}(0, V_{-i})$. Then, we directly compare the utility under truthful report $u_i(\boldsymbol{v}_i, (\boldsymbol{v}_i, V_{-i}))$ and the utility when reporting $\boldsymbol{b}_i$, $u_i(\boldsymbol{v}_i, (\boldsymbol{b}_i, V_{-i}))$. For simplicity, let $V_{-i} = (\boldsymbol{v}_1, \cdots, \boldsymbol{v}_{i-1}, \boldsymbol{v}_{i+1}, \cdots, \boldsymbol{v}_n)$.

$$
\begin{aligned}
u_i(\boldsymbol{v}_i, (\boldsymbol{v}_i, V_{-i})) &= u_i^{\mathrm{AMA}}(\boldsymbol{v}_i, (\boldsymbol{v}_i, V_{-i})) - p_i^{\mathrm{Cor}}(V_{-i}) \\
&\overset{(a)}{\geq} u_i^{\mathrm{AMA}}(\boldsymbol{v}_i, (\boldsymbol{b}_i, V_{-i})) - p_i^{\mathrm{Cor}}(V_{-i}) = u_i(\boldsymbol{v}_i, (\boldsymbol{b}_i, V_{-i})).
\end{aligned}
$$

The inequality (a) is from the DSIC property of AMA.

$\qquad\square$

**Theorem B.1** (The first part of Theorem 3.3)**.** *In single-item auctions, for any number of bidders $n$: If $\mathcal{F}$ is bidder-independent, then $REV_{\mathcal{F}}^{D\text{-}CA} = REV_{\mathcal{F}}^{D\text{-}AMA}$.*

*Proof.* As bidders are independent, we assume that each valuation $\inf\{v_i : v_i \in \mathrm{supp}(\mathcal{F}_i)\} = l_i$ for each $i \in [n]$. We show that when fixing $\mathcal{A}$ to be set of all deterministic allocations, for an optimal solution $(\boldsymbol{w}, \boldsymbol{\lambda}, (p_i^{Cor})_{i=1}^n)$ of Problem CA-AMA-

*OPT, we can construct a feasible solution for Problem AMA-OPT which brings at least the same revenue.* This is sufficient to say that the $\text{REV}^{\text{D-AMA}} \geq \text{REV}^{\text{D-CA}}$.

Let $\mathcal{A}$ be $\{A_0, A_1, A_2, \cdots, A_n\}$, where $A_i$ is the outcome that allocates the item to bidder $i$ and $A_0$ is the outcome that reserves the item. The optimal solution of the CA-AMA is given by $(\boldsymbol{w}, \boldsymbol{\lambda}, (p_i^{\text{Cor}})_{i=1}^n)$. Consider two cases:

If for any $i$ and $\boldsymbol{v}_{-i}$, there is $p_i^{\text{Cor}}(\boldsymbol{v}_{-i}) = 0$, then the revenue of the CA-AMA is equal to the revenue from the AMA parameterized by $(\boldsymbol{w}, \boldsymbol{\lambda})$. *Therefore, below we consider the case that there is at least one $i^*$ and $\boldsymbol{v}_{-i^*}$, such that $p_{i^*}^{\text{Cor}}(\boldsymbol{v}_{-i^*}) > 0$.*

Firstly, the condition $p_{i^*}^{\text{Cor}}(\boldsymbol{v}_{-i^*}) > 0$ means that $g(l_{i^*}, \boldsymbol{v}_{-i^*}; \boldsymbol{w}, \boldsymbol{\lambda}) = A_{i^*}$. Otherwise, the utility of bidder $i^*$ when it realizes its least valuation $l_{i^*}$ is negative, violating the IR constraint. From $g(l_{i^*}, \boldsymbol{v}_{-i^*}; \boldsymbol{w}, \boldsymbol{\lambda}) = A_{i^*}$, we can get the following condition:

$$w_{i^*} l_{i^*} + \lambda_{i^*} > \max\{\max_{j \neq i^*} w_j v_j + \lambda_j, \lambda_0\} \geq \max\{\max_{j \neq i^*} w_j l_j + \lambda_j, \lambda_0\} \geq \lambda_0.$$

Note that this also implies that for any $j \neq i^*$, $p_j^{\text{Cor}}(\boldsymbol{v}_{-j}) \equiv 0$ for any $\boldsymbol{v}_{-j}$. Otherwise, we have $w_{i^*} l_{i^*} + \lambda_{i^*} > w_j l_j + \lambda_j$ and $w_{i^*} l_{i^*} + \lambda_{i^*} < w_j l_j + \lambda_j$ simultaneously.

Secondly, we construct a new AMA based on $(\boldsymbol{w}, \boldsymbol{\lambda})$. Without loss of generality, we set $\lambda_0 = 0$ and define $b := w_{i^*} l_{i^*} + \lambda_{i^*} - \lambda_0 > 0$. The new parameters $(\boldsymbol{w}', \boldsymbol{\lambda}')$ is conducted as $\boldsymbol{w}' = \boldsymbol{w}$, $\lambda_i' = \lambda_i - b$ for all $i \in \{1, 2, \cdots, n\}$, and $\lambda_0' = \lambda_0$.

We analyze the revenue brought by AMA with parameters $(\boldsymbol{w}', \boldsymbol{\lambda}')$. Our goal is to show that for any $\boldsymbol{v} \in \text{supp}(\mathcal{F})$, the payment of the AMA parameterized by $(\boldsymbol{w}', \boldsymbol{\lambda}')$ is at least the payment of the CA-AMA parameterized by $(\boldsymbol{w}, \boldsymbol{\lambda}, (p_i^{\text{Cor}})_{i=1}^n)$.

For any $\boldsymbol{v}$, we obverse that

$$\max_j w_j' v_j + \lambda_j' \geq w_{i^*} v_{i^*} + \lambda_{i^*}' \geq w_{i^*} l_{i^*} + \lambda_{i^*}' = w_{i^*} l_{i^*} + \lambda_{i^*} - b = \lambda_0.$$

*Therefore, the item will always be allocated in the new AMA. Furthermore, as the boost variable $\lambda$ other than $A_0$ changes to the same value, the allocation remains the same.* For this $\boldsymbol{v}$, we consider two cases.

1. The item is allocated to bidder $j \neq i^*$.

As $p_j^{\text{Cor}} = 0$, the original revenue comes solely from $p_j^{\text{AMA}}$. In new AMA mechanism, the $p_j^{\text{AMA}}$ is computed by:

$$\begin{aligned}
w_j' \, p_j^{\text{AMA}}(\boldsymbol{v}; \boldsymbol{w}', \boldsymbol{\lambda}') &= \max\{A_0, \max_{k \neq j} w_k' v_k + \lambda_k'\} - \lambda_j' \\
&= \max\{A_0, \max_{k \neq j} w_k v_k + \lambda_k - b\} - \lambda_j + b \\
&\geq \max\{A_0, \max_{k \neq j} w_k v_k + \lambda_k\} - b - \lambda_j + b \\
&= \max\{A_0, \max_{k \neq j} w_k v_k + \lambda_k\} - \lambda_j \\
&= w_j \, p_j^{\text{AMA}}(\boldsymbol{v}; \boldsymbol{w}, \boldsymbol{\lambda}) = w_j' \, p_j^{\text{AMA}}(\boldsymbol{v}; \boldsymbol{w}, \boldsymbol{\lambda}).
\end{aligned}$$

2. The item is allocated to bidder $i^*$.

We compare the revenue between $p_{i^*}^{\text{AMA}}(\boldsymbol{v}; \boldsymbol{w}', \boldsymbol{\lambda}')$ and $p_{i^*}^{\text{AMA}}(\boldsymbol{v}; \boldsymbol{w}, \boldsymbol{\lambda}) + p_i^{\text{Cor}}(\boldsymbol{v}_{-i^*})$. Firstly,

$$\begin{aligned}
w_{i^*}' \, p_{i^*}^{\text{AMA}}(\boldsymbol{v}; \boldsymbol{w}', \boldsymbol{\lambda}') &= \max\{A_0, \max_{k \neq i^*} w_k' v_k + \lambda_k'\} - \lambda_{i^*}' \\
&= \max\{A_0, \max_{k \neq i^*} w_k v_k + \lambda_k - b\} - \lambda_{i^*} + b \\
&\geq \lambda_0 - \lambda_{i^*} + b \\
&= w_{i^*} l_{i^*} + \lambda_{i^*} - \lambda_0 + \lambda_0 - \lambda_{i^*} \\
&= w_{i^*} l_{i^*}.
\end{aligned}$$

For $p_{i^*}^{\mathrm{Cor}}(\boldsymbol{v}_{-i^*})$, by IR constraint, we have,

$$p_{i^*}^{\mathrm{Cor}}(\boldsymbol{v}_{-i^*}) \leq l_{i^*} - p_{i^*}^{\mathrm{AMA}}(l_{i^*}, \boldsymbol{v}_{-i^*}; \boldsymbol{w}, \boldsymbol{\lambda})$$
$$= l_{i^*} - \max\{A_0, \max_{k \neq i^*} w_k v_k + \lambda_k\} + \lambda_{i^*}$$
$$= l_{i^*} - p_{i^*}^{\mathrm{AMA}}(\boldsymbol{v}; \boldsymbol{w}, \boldsymbol{\lambda}).$$

Therefore, $p_{i^*}^{\mathrm{Cor}}(\boldsymbol{v}_{-i^*}) + p_{i^*}^{\mathrm{AMA}}(\boldsymbol{v}; \boldsymbol{w}, \boldsymbol{\lambda}) \leq l_{i^*} \leq p_{i^*}^{\mathrm{AMA}}(\boldsymbol{v}; \boldsymbol{w}', \boldsymbol{\lambda}')$.

Hence, for any valuation profile $\boldsymbol{w}$, the revenue by AMA $(\boldsymbol{w}', \boldsymbol{\lambda}')$ is at least the revenue given by CA-AMA $(\boldsymbol{w}, \boldsymbol{\lambda}, (p_i^{\mathrm{Cor}})_{i=1}^n)$.
□

**Theorem B.2** (The second part of Theorem 3.3)**.** *In single-item auctions, for any number of bidders $n$ and any $\epsilon > 0$, there exists a distribution $\mathcal{F}$ such that $REV_{\mathcal{F}}^{\mathrm{D\text{-}AMA}} \leq \epsilon \cdot REV_{\mathcal{F}}$, while $REV_{\mathcal{F}}^{\mathrm{D\text{-}CA}} = REV_{\mathcal{F}}$. Furthermore, $REV_{\mathcal{F}}^{\mathrm{S\text{-}AMA}} < REV_{\mathcal{F}}$ for any $S$.*

*Proof.* The valuation distribution for the single-item auction is set as follows: Bidder 1's valuation follows a equal revenue distribution on $[\epsilon, 1]$, i.e., the pdf is given by $f(v) = \frac{\epsilon}{(1-\epsilon)v^2}$. The other bidders' valuations are the same and are linear to $v_1$, $v_i = \epsilon_1 \cdot (1 - v_1)$, for all $i \geq 2$. We require $0 < \epsilon_1 < \epsilon < 1$, with specific values to be determined later.

Part 1: Showing $REV_{\mathcal{F}} = REV_{\mathcal{F}}^{\mathrm{D\text{-}CA}}$.

For this distribution, it is possible to extract the full social surplus $\max_{i \in [n]} v_i$ as payment for every valuation profile $\boldsymbol{v}$. In the CA-AMA framework, we achieve this by setting: $p_1^{\mathrm{Cor}}(\boldsymbol{v}_{-1}) = (1 - v_2/\epsilon)$, $\mathcal{A}$ to be set of all deterministic allocation, $\boldsymbol{w} = 1$, $\boldsymbol{\lambda}_k = 0$ for all $k \in [S]$. By this, the revenue is the same as first-price auction:

$$REV_{\mathcal{F}} = REV_{\mathcal{F}}^{\mathrm{CA}} = \int_{\epsilon}^{1} f(v)v \, \mathrm{d}v = \int_{\epsilon}^{1} \frac{\epsilon}{(1-\epsilon)v} \, \mathrm{d}v = \frac{\epsilon \ln(1/\epsilon)}{1 - \epsilon}.$$

Part 2: Showing the relationship between $REV_{\mathcal{F}}^{\mathrm{D\text{-}AMA}}$ and $REV_{\mathcal{F}}$.

In deterministic AMA, $\mathcal{A}$ is fixed to be $\{A_0, A_1, A_2, \cdots, A_n\}$, where $A_i$ is the outcome that allocates the item to bidder $i$ and $A_0$ is the outcome that reserves the item. We first show the following lemma:

**Lemma B.3.** *Under the constructed valuation, for any bidder $1$'s valuation $v < v'$ and AMA parameter $(\boldsymbol{w}, \boldsymbol{\lambda})$, if bidder $1$ wins the item on $v$, then it also wins the item on $v'$.*

*Proof.* When bidder 1's valuation is $v$ and wins the item, we have:

$$w_1 v + \lambda_1 \geq \max\{\lambda_0, \max_{j \geq 2} w_j v_j + \lambda_j\} = \max\{\lambda_0, \max_{j \geq 2} w_j \epsilon_1 (1 - v) + \lambda_j\}.$$

Then for any valuation $v' > v$, we still have that:

$$w_1 v' + \lambda_1 > w_1 v + \lambda_1$$
$$\geq \max\{\lambda_0, \max_{j \geq 2} w_j \epsilon_1 (1 - v) + \lambda_j\}$$
$$\geq \max\{\lambda_0, \max_{j \geq 2} w_j \epsilon_1 (1 - v') + \lambda_j\}.$$

This means that bidder 1 will also win the item.
□

We consider two cases: (1) Bidder 1 never wins the item: then the payment will always be lower than the valuation of other bidders and hence is at most $\epsilon_1$. (2) Bidder 1 does not win when its valuation is less than $v^*$ and wins when its valuation is in $[v^*, 1]$. Still, the payment collected when its valuation is less than $v^*$ is at most $\epsilon_1 \int_{\epsilon}^{v^*} f(v)dv \leq \epsilon_1$. The payment for the bidder 1 when it wins is bounded by

$$p_1^{\mathrm{AMA}}(\boldsymbol{v}; \boldsymbol{w}, \boldsymbol{\lambda}) = \frac{1}{w_1} \left( \max\{\lambda_0, \lambda_1, \max_{j \geq 2} w_j \epsilon_1 (1 - v) + \lambda_j\} - \lambda_1 \right)$$
$$\leq \frac{1}{w_1} \left( \max\{\lambda_0, \lambda_1, \max_{j \geq 2} w_j \epsilon_1 (1 - v^*) + \lambda_j\} - \lambda_1 \right)$$
$$= p_1^{\mathrm{AMA}}((v^*, \epsilon_1(1 - \boldsymbol{v}^*)); \boldsymbol{w}, \boldsymbol{\lambda}) \leq v^*.$$

The last inequality is derived from the IR property of any AMA. Therefore, the upper bound of the payment for $[v^*, 1]$ can be computed by:

$$\int_{v^*}^1 v^* f(v) dv = \int_{v^*}^1 v^* \frac{\epsilon}{(1-\epsilon)v^2} dv = v^* \frac{\epsilon}{(1-\epsilon)} \left( \frac{1}{v^*} - 1 \right) \leq \frac{\epsilon}{(1-\epsilon)}.$$

Therefore, the expected payment is bounded by $\frac{\epsilon}{(1-\epsilon)} + \epsilon_1$. As $\text{REV}_{\mathcal{F}} = \frac{\epsilon \ln(1/\epsilon)}{1-\epsilon}$. For any $\delta$, we can easily set $\epsilon$ and $\epsilon_1$ so that $\text{REV}_{\mathcal{F}}^{\text{D-AMA}} < \delta \cdot \text{REV}_{\mathcal{F}}$.

Part 3: Showing the relationship between $\text{REV}_{\mathcal{F}}^{\text{S-AMA}}$ and $\text{REV}_{\mathcal{F}}$.

As we consider the case that the size of the allocation menu is finite, i.e., $|\mathcal{A}| = S$, $S$ is a constant. Denote the winning allocation as a function to $v_1$, $k(v_1) = \arg\max_{k \in [S]} w_1 v_1 (A_k)_1 + \sum_{j \geq 2} w_j v_j (A_k)_j + \lambda_k = \arg\max_{k \in [S]} w_1 v_1 (A_k)_1 + \sum_{j \geq 2} w_j \epsilon_1 (1-v_1)(A_k)_j + \lambda_k$. The function must be a piece-wise constant function, and the function has at most $S$ change points by the following lemma.

**Lemma B.4.** *For any $(\mathcal{A}, \boldsymbol{w}, \boldsymbol{\lambda})$, there is at most $S = |\mathcal{A}|$ change points of $k(v_1)$.*

*Proof.* We prove this result by contradiction. Assume that there are $S + 1$ change points, then, there must be a case that for $v_1^1 < v_1^2 < v_1^3$ such that $k = g(v_1^1) = g(v_1^3)$, $k' = g(v_1^2)$, and $k \neq k'$. Then, by definition of AMA's allocation rule, we have

$$w_1 v_1^1 (A_k)_1 + \sum_{j \geq 2} w_j v_j^1 (A_k)_j + \lambda_k \geq w_1 v_1^1 (A_{k'})_1 + \sum_{j \geq 2} w_j v_j^1 (A_{k'})_j + \lambda_{k'} \tag{a}$$

$$w_1 v_1^2 (A_{k'})_1 + \sum_{j \geq 2} w_j v_j^2 (A_{k'})_j + \lambda_{k'} \geq w_1 v_1^2 (A_k)_1 + \sum_{j \geq 2} w_j v_j^2 (A_k)_j + \lambda_k \tag{b}$$

$$w_1 v_1^3 (A_k)_1 + \sum_{j \geq 2} w_j v_j^3 (A_k)_j + \lambda_k \geq w_1 v_1^3 (A_{k'})_1 + \sum_{j \geq 2} w_j v_j^3 (A_{k'})_j + \lambda_{k'}. \tag{c}$$

Inserting $v_j = \epsilon_1 (1 - v_1) \quad \forall j \geq 2$, by (b) - (a), we have $w_1((A_{k'})_1 - (A_k)_1) \geq \epsilon_1 \sum_{j \geq 2} w_j((A_{k'})_j - (A_k)_j)$. By (c) - (b), we have $w_1((A_k)_1 - (A_{k'})_1) \geq \epsilon_1 \sum_{j \geq 2} w_j((A_k)_j - (A_{k'})_j)$. The only feasible solution is that $(A_k)_j = (A_{k'})_j$ for all $j \in [n]$, which means $A_k = A_{k'}$ and hence brings a contradiction. $\square$

Therefore, we know that there are at most $S$ change points of $g(v_1)$. Suppose these $S' \leq S$ change points are $v_1^0 = \epsilon < v_1^1 < v_1^2 < \cdots < v_1^{S'} < v_1^{S'+1} = 1$ and the corresponding allocations are $A_0, A_1, A_2, \cdots, A_{S'}$. We only consider the interval $[v_1^0, v_1^1)$. If in this interval, $(A_0)_1 < 1$, which means the item is not allocated to bidder 1 deterministically, then the payment loss compared to optimal revenue is at least $(1 - (A_0)_1) \int_{v_1^0 = \epsilon}^{v_1^1} (v - \epsilon_1) dv > 0$.

On the other hand, if the allocation satisfies that $(A_0)_1 = 1$. From a similar proof above, we know that the payment in this interval is at most $v_1^0$, which will also results in a gap of $\int_{v_1^0 = \epsilon}^{v_1^1} (v - v_1^0) f(v) dv > 0$ compared to the optimal revenue. Therefore, in both cases, we can induce that $\text{REV}_{\mathcal{F}}^{\text{S-AMA}} < \text{REV}_{\mathcal{F}}$. $\square$

## C. Omitted Proofs in Section 4

**Theorem 4.1.** *The target function $p_i^{\text{OPT-core}}$ is continuous with respect to the AMA parameters $\mathcal{A}$, $\boldsymbol{w}$, and $\boldsymbol{\lambda}$. Furthermore, assume that there exists a constant $C_H > 0$ such that for all $V_{-i}, V'_{-i}$, the Hausdorff distance $h(supp(\mathcal{F}_i(V_{-i})), supp(\mathcal{F}_i(V'_{-i}))) \leq C_H \|V_{-i} - V'_{-i}\|$, then $p_i^{\text{OPT-core}}$ is also continuous with respect to $V_{-i}$.*

*Proof.* For simplicity, we use $\phi$ to represent AMA parameters $(\mathcal{A}, \boldsymbol{w}, \boldsymbol{\lambda})$. Specifically, $\mathcal{A} = \{A_1, A_2, \cdots, A_S\}$, $\boldsymbol{w} = \{w_1, w_2, \cdots, w_n\}$, and $\boldsymbol{\lambda} = \{\lambda_1, \lambda_2, \cdots, \lambda_S\}$. *For any matrices $A$, $A'$ (vectors $v$, $v'$), we denote notation $d_1(A, A')$ ($d_1(v, v')$) the $L_1$ distance.* For two $\phi$ and $\phi'$, denote $d_1(\phi, \phi') = \sum_{k=1}^S d_1(A_k, A'_k) + d_1(\boldsymbol{w}, \boldsymbol{w}') + d_1(\boldsymbol{\lambda}, \boldsymbol{\lambda}')$.

Recall that $\text{asw}(k; V, \phi)$ is the affine social welfare given by the $k$-th allocation in $\mathcal{A}$, which means: $\text{asw}(k; V, \phi) = \sum_{j=1}^n w_j(v_j \cdot (A_k)_j) + \lambda_k$. We first show that $\text{asw}(k; V, \phi)$ is continuous w.r.t $\phi$. For any $\phi, \epsilon, \phi'$ such that $d_1(\phi, \phi') \leq \epsilon$,

and $k \in [S]$, let $\bar{w} := \max_j w_j$, we have

$$|\mathrm{asw}(k; V, \phi) - \mathrm{asw}(k; V, \phi')|$$

$$= |\sum_{j=1}^n w_j(v_j \cdot (A_k)_j) + \lambda_k - \sum_{j=1}^n w_j'(v_j \cdot (A_k')_j) - \lambda_k'|$$

$$= |\sum_{j=1}^n w_j(v_j \cdot (A_k)_j) - \sum_{j=1}^n w_j(v_j \cdot (A_k')_j)$$

$$+ \sum_{j=1}^n w_j(v_j \cdot (A_k')_j) - \sum_{j=1}^n w_j'(v_j \cdot (A_k')_j) + \lambda_k - \lambda_k'|$$

$$\leq \sum_{j=1}^n w_j v_j \cdot ((A_k)_j - (A_k')_j) + \sum_{j=1}^n |w_j - w_j'|(v_j \cdot (A_k')_j) + |\lambda_k - \lambda_k'|$$

$$\leq \bar{w} \sum_{j=1}^n d_1(A_k, A_k') + m d_1(\boldsymbol{w}, \boldsymbol{w}') + d_1(\boldsymbol{\lambda}, \boldsymbol{\lambda}')$$

$$\leq \max\{\bar{w}, m\} d_1(\phi, \phi') \leq \max\{\bar{w}, m\} \epsilon.$$

This means that the continuity of $\phi$ holds.

**(1) The continuity with respect to AMA parameters $\phi$.**

We use asw to compute a bidder's utility under AMA. By the allocation rule and payment rule defined by AMA, there is

$$u_i^{\mathrm{AMA}}(\boldsymbol{v}_i, V; \phi) = \frac{1}{w_i}\left(\max_{k \in [S]} \mathrm{asw}(k; V, \phi) - \max_{k \in [S]} \mathrm{asw}(k; (0, V_{-i}), \phi)\right).$$

And for the target function,

$$p_i^{\mathrm{OPT-Cor}}(V_{-i}; \phi) = \inf_{\boldsymbol{v}_i \in \mathrm{supp}(\mathcal{F}_i(V_{-i}))} u_i^{\mathrm{AMA}}(\boldsymbol{v}_i, (\boldsymbol{v}_i, V_{-i}); \phi).$$

As both max and inf operations do not influence the continuity, we can conclude that $p_i^{\mathrm{OPT-Cor}}(V_{-i}; \phi)$ is continuous w.r.t. $\phi$ for any $V_{-i}$.

**(2) Continuity in the other bidders' valuations $V_{-i}$.**

Here, the AMA parameters $\phi$ are fixed; we first show that asw is also continuous to $V$. For any $\phi$, $k$, $V$ and $V'$, we have

$$|\mathrm{asw}(k; V, \phi) - \mathrm{asw}(k; V, \phi)|$$

$$= |\sum_{j=1}^n w_j(\boldsymbol{v}_j \cdot (A_k)_j) + \lambda_k - \sum_{j=1}^n w_j(\boldsymbol{v}_j' \cdot (A_k)_j) - \lambda_k|$$

$$= |\sum_{j=1}^n w_j(\boldsymbol{v}_j \cdot (A_k)_j) - \sum_{j=1}^n w_j(\boldsymbol{v}_j' \cdot (A_k)_j)|$$

$$\leq \sum_{j=1}^n w_j(|\boldsymbol{v}_j - \boldsymbol{v}_j'| \cdot (A_k)_j) = \sum_{j=1}^n w_j \sum_{t=1}^m |\boldsymbol{v}_{jt} - \boldsymbol{v}_{jt}'|(A_k)_{jt}$$

$$= \sum_{t=1}^m \sum_{j=1}^n w_j |\boldsymbol{v}_{jt} - \boldsymbol{v}_{jt}'|(A_k)_{jt} \leq \sum_{t=1}^m \max_j w_j |\boldsymbol{v}_{jt} - \boldsymbol{v}_{jt}'| \leq \bar{w} d_1(V, V').$$

Then, as the mechanism satisfies DSIC, we will use notation $u_i^{\mathrm{AMA}}(\boldsymbol{v}_i, V_{-i}; \phi)$ to represent the original $u_i^{\mathrm{AMA}}(\boldsymbol{v}_i, (\boldsymbol{v}_i, V_{-i}); \phi)$ as bidders' will always truthfully report. As $u_i^{\mathrm{AMA}}$ is a maximum of a finite number of continuous functions, for any $\boldsymbol{v}_i$, $\boldsymbol{v}_i'$, $V_{-i}$ and $V_{-i}'$, let $L = \frac{2\bar{w}}{w_i}$, we have

$$|u_i^{\mathrm{AMA}}(\boldsymbol{v}_i, V_{-i}; \phi) - u_i^{\mathrm{AMA}}(\boldsymbol{v}_i', V_{-i}'; \phi)| \leq L d_1(\boldsymbol{v}_i, \boldsymbol{v}_i') + L d_1(V_{-i}, V_{-i}').$$

Now, for two valuation profiles $V_{-i}$, $V'_{-i}$, by definition of $p_i^{\text{OPT}-\text{Cor}}$, for any $\epsilon > 0$, we can find a $\boldsymbol{v}_i \in \text{supp}\mathcal{F}_i(V_{-i})$ such that $p_i^{\text{OPT}-\text{Cor}}(V_{-i}) \leq u_i^{\text{AMA}}(\boldsymbol{v}_i, V_{-i}; \phi) \leq p_i^{\text{OPT}-\text{Cor}}(V_{-i}) + \epsilon$. By the Hausdorff assumption on $\text{supp}\mathcal{F}_i(V_{-i})$ and $\text{supp}\mathcal{F}_i(V'_{-i})$, we can find another $\boldsymbol{v}'_i \in \text{supp}\mathcal{F}_i(V'_{-i})$, such that $d_1(\boldsymbol{v}_i, \boldsymbol{v}'_i) \leq C_H d_1(V_{-i}, V'_{-i})$. Therefore, we can bound the gap in the values

$$
\begin{aligned}
p_i^{\text{OPT}-\text{Cor}}(V_{-i}; \phi) &\geq u_i^{\text{AMA}}\big(\boldsymbol{v}_i, (\boldsymbol{v}_i, V_{-i}); \phi\big) - \epsilon \\
&\geq u_i^{\text{AMA}}\big(\boldsymbol{v}'_i, (\boldsymbol{v}'_i, V'_{-i}); \phi\big) - L\, d_1(\boldsymbol{v}_i, \boldsymbol{v}'_i) - L\, d_1(V_{-i}, V'_{-i}) - \epsilon \\
&\geq u_i^{\text{AMA}}\big(\boldsymbol{v}'_i, (\boldsymbol{v}'_i, V'_{-i}); \phi\big) - L(C_H + 1)\, d_1(V_{-i}, V'_{-i}) - \epsilon \\
&\geq p_i^{\text{OPT}-\text{Cor}}(V'_{-i}; \phi) - \epsilon - L(C_H + 1)\, d_1(V_{-i}, V'_{-i}).
\end{aligned}
$$

It is obvious that the vice is also correct, so we can conclude that:

$$
\begin{aligned}
&|p_i^{\text{OPT}-\text{Cor}}(V_{-i}; \phi) - p_i^{\text{OPT}-\text{Cor}}(V'_{-i}; \phi)| \\
&\leq \epsilon + L(C_H + 1)\, d_1(V_{-i}, V'_{-i}) = \epsilon + \frac{2\bar{w}}{w_i}(C_H + 1)\, d_1(V_{-i}, V'_{-i}).
\end{aligned}
$$

As $\epsilon$ can be chosen sufficiently small, this means that $p_i^{\text{OPT}-\text{Cor}}(\cdot; \phi)$ is $\frac{2\bar{w}}{w_i}(C_H + 1)$-continuous w.r.t. $V_{-i}$ under $L_1$ distance for any fixed $\phi$ under $C_H$-Hausdorff assumption. $\qquad\square$

**Theorem C.1** (Uniform generalization bound for a 3-layer payment network). *Let $\mathcal{F}$ be an arbitrary distribution over valuation profiles $V \in [0, 1]^{n \times m}$. For parameters $\theta = (W_1, W_2, W_3)$ satisfying $\|W_\ell\|_2 \leq M_\ell$ for $\ell = 1, 2, 3$, consider $Regret_{IR}(V) = \sum_{i=1}^n \max\{0, p_i^{Cor}(V_{-i}; \theta) - u_i^{AMA}(\boldsymbol{v}_i, V)\}$, where the payment network $p_i^{Cor}(\,\cdot\,; \theta) : \mathbb{R}^{(n-1)m} \to \mathbb{R}$ is the depth-3 ReLU network $p_i(x; \theta) = W_3\, \sigma\big(W_2\, \sigma(W_1 x)\big)$ with $\sigma(z) = \max\{0, z\}$.*

*Let $B_x = \sqrt{(n-1)m}$, and $B_p = B_x \prod_{\ell=1}^3 M_\ell$. For any i.i.d. sample $D = \{V^{(1)}, \ldots, V^{(K)}\} \sim \mathcal{F}^K$ and any confidence level $\delta \in (0, 1)$, with probability at least $1 - \delta$ (over the draw of $D$) the following inequality holds simultaneously for every choice of parameters $\theta$:*

$$
\sup_\theta \left| \frac{1}{K} \sum_{k=1}^K Regret_{IR}(V^{(k)}; \theta) - \mathbb{E}\,Regret_{IR}(V; \theta) \right| \leq 2n\frac{B_p\sqrt{2\log(2d)}}{\sqrt{K}} + nB_p\sqrt{\frac{\log(2/\delta)}{2K}},
$$

*where $d = \max\{(n-1)m, h_1, h_2, 1\}$ and $h_1, h_2$ are the widths of the first and second hidden layers.*

*Proof.* Since every valuation component lies in $[0, 1]$, $\|V_{-i}\|_2 \leq B_x = \sqrt{(n-1)m}$. For ReLU networks, the operator norm is non-expansive, hence, $|p_i^{\text{Cor}}(V_{-i}; \theta)| \leq \|W_3\|_2 \|W_2\|_2 \|W_1\|_2 \|V_{-i}\|_2 \leq B_p$. Together with $0 \leq u_i(\boldsymbol{v}_i, V) \leq m$ we therefore have $0 \leq \text{Regret}_{\text{IR}}(V; \theta) \leq nB_p$.

Let $\mathcal{P} = \{\text{Regret}_{\text{IR}}(V; \theta) : \theta \in \Theta\}$. By standard symmetrisation (see, e.g., *Bartlett & Mendelson, 2002*), for any fixed sample $D$

$$
\sup_\theta \left| \frac{1}{K} \sum_{k=1}^K \text{Regret}_{\text{IR}}(V^{(k)}; \theta) - \mathbb{E}_{V \sim \mathcal{F}}[\text{Regret}_{\text{IR}}(V; \theta)] \right| \leq 2\,\widehat{R}_K(\mathcal{P}) + nB_p\sqrt{\frac{\log(2/\delta)}{2K}}
$$

with probability $\geq 1 - \delta$, where $\widehat{R}_K$ is the empirical Rademacher complexity.

Let $\mathcal{P}_i = \{p_i^{\text{Cor}}(V_{-i}; \theta) : \theta \in \Theta\}$ be the function class for a single payment component. For a depth-3 ReLU network with spectral-norm bounds $M_\ell$, we have

$$
\widehat{R}_K(\mathcal{P}_i) \leq \frac{B_x\big(\prod_{\ell=1}^3 M_\ell\big)\sqrt{2\log(2d)}}{\sqrt{K}},
$$

where $d = \max\{(n-1)m, h_1, h_2, 1\}$ and $h_1, h_2$ are the widths of the first and second hidden layers.

Since $\text{Regret}_{\text{IR}}(V; \theta) = \sum_{i=1}^n \max\{0, p_i^{\text{Cor}}(V_{-i}; \theta) - u_i^{\text{AMA}}(\boldsymbol{v}_i, V)\}$ and $\max\{0, \cdot\}$ is 1-Lipschitz, we have:

$$
\widehat{R}_K(\mathcal{P}) \leq \sum_{i=1}^n \widehat{R}_K(\{p_i^{\text{Cor}}(V_{-i}; \theta)\}) = n \cdot \widehat{R}_K(\mathcal{P}_i).
$$

Substituting the bound for $\widehat{R}_K(\mathcal{P}_i)$:

$$\widehat{R}_K(\mathcal{P}) \;\leq\; n\frac{B_p\sqrt{2\log(2d)}}{\sqrt{K}}.$$

Finally, substituting the complexity estimate for $\widehat{R}_K(\mathcal{P})$ finishes the proof. □

*Remark* C.2 (Fixed network). If $\theta$ is treated as *fixed* (e.g. after training), *Hoeffding's inequality* immediately gives the simpler bound

$$\left| \frac{1}{K}\sum_k f_{i,\theta}(V^{(k)}) - \mathbb{E}f_{i,\theta}(V) \right| \leq nB_p\sqrt{\frac{\log(2/\delta)}{2K}},$$

so the capacity term vanishes.

# D. Algorithm of CA-AMA

We present the detailed algorithm description for classic randomized AMA optimization methods, including Lottery­AMA (Curry et al., 2023) and AMenuNet (Duan et al., 2023) in Algorithm 1. For the softmax version of AMA, given a valuation profile $V$, the AMA parameters $(\mathcal{A}, \boldsymbol{w}, \boldsymbol{\lambda})$ and temperature $T$, the approximated allocation is calculated as follows,

$$\hat{g}^{\mathrm{AMA}}(V) = \sum_{A\in\mathcal{A}} \frac{e^{\mathrm{asw}(A;V)\cdot T}}{\sum_{A'\in\mathcal{A}} e^{\mathrm{asw}(A';V)\cdot T}} A,$$

$$\hat{g}^{\mathrm{AMA}}_{-i}(V) = \sum_{A\in\mathcal{A}} \frac{e^{\mathrm{asw}_{-i}(A;V)\cdot T}}{\sum_{A'\in\mathcal{A}} e^{\mathrm{asw}_{-i}(A';V)\cdot T}} A.$$

$\mathrm{asw}(k;V)$ is defined as $\sum_{j=1}^{n} w_j \boldsymbol{v}_j \cdot (A_k)_j + \lambda_k$ and $\mathrm{asw}_{-i}(k;V)$ is $\sum_{j=1,j\neq i}^{n} w_j \boldsymbol{v}_j \cdot (A_k)_j + \lambda_k$. Based on that, the payment and utility for bidder $i$ is:

$$
\begin{aligned}
\hat{p}^{\mathrm{AMA}}_i(V) &= \frac{1}{w_i}\left( \mathrm{asw}_{-i}(\hat{g}^{\mathrm{AMA}}_{-i}(V); V) - \mathrm{asw}_{-i}(\hat{g}^{\mathrm{AMA}}(V); V) \right), \\
\hat{u}^{\mathrm{AMA}}_i(V) &= \boldsymbol{v}_i \cdot \hat{g}^{\mathrm{AMA}}(V)_i - \hat{p}^{\mathrm{AMA}}_i(V).
\end{aligned}
\tag{3}
$$

Note that in this approximated version, all operations are differentiable to the AMA parameters $(\mathcal{A}, \boldsymbol{w}, \boldsymbol{\lambda})$. For other notations and equations, please refer to Section 4.

# E. Comparison with Fully Expressive Baselines

We provide a detailed qualitative comparison between CA-AMA and fully expressive baseline mechanisms, particularly GemNet (Wang et al., 2024) and RegretNet (Dütting et al., 2023), to clarify the positioning of our work within the broader landscape of differentiable mechanism design.

**CA-AMA as a Menu-Based Mechanism.** CA-AMA can be interpreted as a specialized menu-based mechanism. When the ex-post opt-out option is introduced, CA-AMA essentially offers each bidder a menu with exactly **two options**: (1) the allocation and payment derived by the CA-AMA network, or (2) zero allocation for zero payment. While general menu-based mechanisms (e.g., GemNet) can theoretically express any DSIC and IR mechanism, CA-AMA's simplified two-option structure means it may not capture the absolute optimal mechanism across all possible distributions.

**Practical Advantages over Fully Expressive Mechanisms.** We emphasize two crucial practical advantages that CA-AMA holds over mechanisms with full theoretical expressiveness:

1. **Versus RegretNet:** (Dütting et al., 2023) Because CA-AMA intrinsically guarantees DSIC by construction, it entirely avoids the computationally expensive process of calculating IC-regret during training. RegretNet requires sampling alternative bids to approximate incentive violations, which incurs significant computational overhead.

---

**Algorithm 1** Unified CA-AMA Optimization Framework

---

**Require:** Data generator $\mathcal{G}$, initial parameters $\theta$ (and optionally $\phi$), total iterations $T$, sample size $K$, training mode
  mode $\in \{\texttt{baseline}, \texttt{mutual}, \texttt{post}\}$,
     (if mode $\neq$ baseline): hyperparameters $\gamma, \gamma_\Delta, R_{\text{target}}, \bar{\gamma}$.
 1: Initialize neural network $p^\theta$ (AMA parameters).
 2: **if** mode $\neq$ baseline **then**
 3:    Initialize neural network $p^\phi$ (correlation-aware payments).
 4:    Set initial penalty strength $\gamma$.
 5: **end if**
 6: **for** $t = 1$ to $T$ **do**
 7:    Generate dataset $D = \{V^{(1)}, V^{(2)}, \ldots, V^{(K)}\}$ by $\mathcal{G}$.
 8:    Get $\mathcal{A}$, $\boldsymbol{w}$, and $\boldsymbol{\lambda}$ from $p^\theta$.
 9:    **for** $i = 1$ to $n$ **do**
10:      **if** mode $=$ post **then**
11:         Compute **exact** AMA payment $p_i^{\text{AMA}}$ and utility $u_i^{\text{AMA}}$ using true $\text{argmax}$.
12:      **else**
13:         Approximate AMA payment $\hat{p}_i^{\text{AMA}}$ and utility $\hat{u}_i^{\text{AMA}}$ using $\text{softmax}$.
14:      **end if**
15:      **if** mode $\neq$ baseline **then**
16:         Get correlation-aware payment $p_i^{\text{Cor}}$ by $p^\phi$.
17:      **end if**
18:    **end for**
19:    Compute loss by Eqn (1)
20:    **if** mode $=$ baseline **then**
21:      Update $p^\theta$ by gradient descent on $\mathcal{L}$.
22:    **else if** mode $=$ post **then**
23:      Freeze $p^\theta$; update only $p^\phi$ by gradient descent on $\mathcal{L}$.
24:    **else**
25:      Update both $p^\theta$ and $p^\phi$ by gradient descent on $\mathcal{L}$.
26:    **end if**
27:    **if** mode $\neq$ baseline **then**
28:      Update penalty strength $\gamma$ with Eqn (2).
29:    **end if**
30: **end for**

---

*Table 3.* Hyperparameters and training times of CA-AMA and Randomized AMA methods.

| Hyperparameter | 2×2 | 5×2 | 8×2 | 10×2 | 2×3 |
|---|---|---|---|---|---|
| Initial penalization term $\gamma_0$ | 3 | 6 | 6 | 8 | 5 |
| Menu size $|\mathcal{A}|$ | 32 | 64 | 128 | 256 | 64 |
| CA-AMA training time (min) | 20 | 26 | 40 | 47 | 22 |
| AMenuNet training time (min) | 19 | 23 | 33 | 40 | 20 |

| Hyperparameter | 5×3 | 8×3 | 10×3 | 2×5 | 5×5 |
|---|---|---|---|---|---|
| Initial penalization term $\gamma_0$ | 6 | 8 | 8 | 3 | 10 |
| Menu size $|\mathcal{A}|$ | 1024 | 2048 | 2048 | 256 | 2048 |
| CA-AMA training time (min) | 40 | 80 | 90 | 27 | 70 |
| AMenuNet training time (min) | 40 | 75 | 85 | 24 | 65 |

2. **Versus GemNet:** (Wang et al., 2024) CA-AMA naturally satisfies strict allocation feasibility without requiring the complex integer programming or post-processing steps necessary in GemNet. GemNet's use of over-allocation penalties and subsequent mixed-integer linear programming for feasibility restoration makes it significantly more difficult to optimize and slower at runtime.

**Positioning on the Pareto Front.** While CA-AMA trades off full theoretical expressiveness, it consistently achieves highly competitive empirical revenue across diverse distributions, as demonstrated in our experiments. Consequently, we view CA-AMA as occupying a valuable position on the Pareto front between theoretical expressiveness, strict incentive guarantees, and empirical performance.

# F. Further Experimental Descriptions

In this section, we presents more implementation details and further experimental results.

## F.1. Implementation Details

As we have introduced in Section 5, most hyperparameters are the same for all settings as our method shows robustness in different auction environments. Only two hyperparameters vary for different settings: the initial penalization term $\gamma_0$ and the menu size $|\mathcal{A}|$. In Table 3, we present the choices taken in our experiments, and the total training time for different auction settings ($n$ and $m$). As the implementation of CA-AMA only adds a computation for the Regret$_{\text{IR}}$ term and the correlation-aware payment is represented by simply a three-layer MLP, the training time does not significantly increase compared to (Duan et al., 2023).

**Generator of AMA Parameters** This part briefly describes how the AMA parameters are over-parameterized by neural networks in our implementation. We refer the reader to AMenuNet (Duan et al., 2023) for more details as our innovation is conceptually orthogonal to the architecture design of optimizing AMA parameters. Specifically, we use some padding as input to a transformer-based neural network to generate an intermediate representation. Then, the allocation matrices $\mathcal{A}$ and the weights $\boldsymbol{w}$ are directly induced by the representation by reshaping and normalization. And the boost variables $\boldsymbol{\lambda}$ are generated by another MLP based on the representation. Empirical results in Duan et al. (2023) show that this over-parameterization technique can significantly improve the performance compared to directly optimizing the AMA parameters.

## F.2. Further Experimental Results

We present more experimental results for the auction setting with linearly correlated equal revenue distributions, as described in Section 5.2. As is shown in Figure 4, the CA-AMA mechanism consistently achieves optimal revenue across different values of $\epsilon$, significantly outperforming the Randomized AMA mechanism implemented by AMenuNet (Duan et al., 2023).

Furthermore, we investigate the impact of the target level of IR regret, $R_{\text{target}}$, on the revenue achieved by our optimized CA-AMA mechanism. Experiments are conducted in a 2-bidder 2-item auction setting with irregular mul-

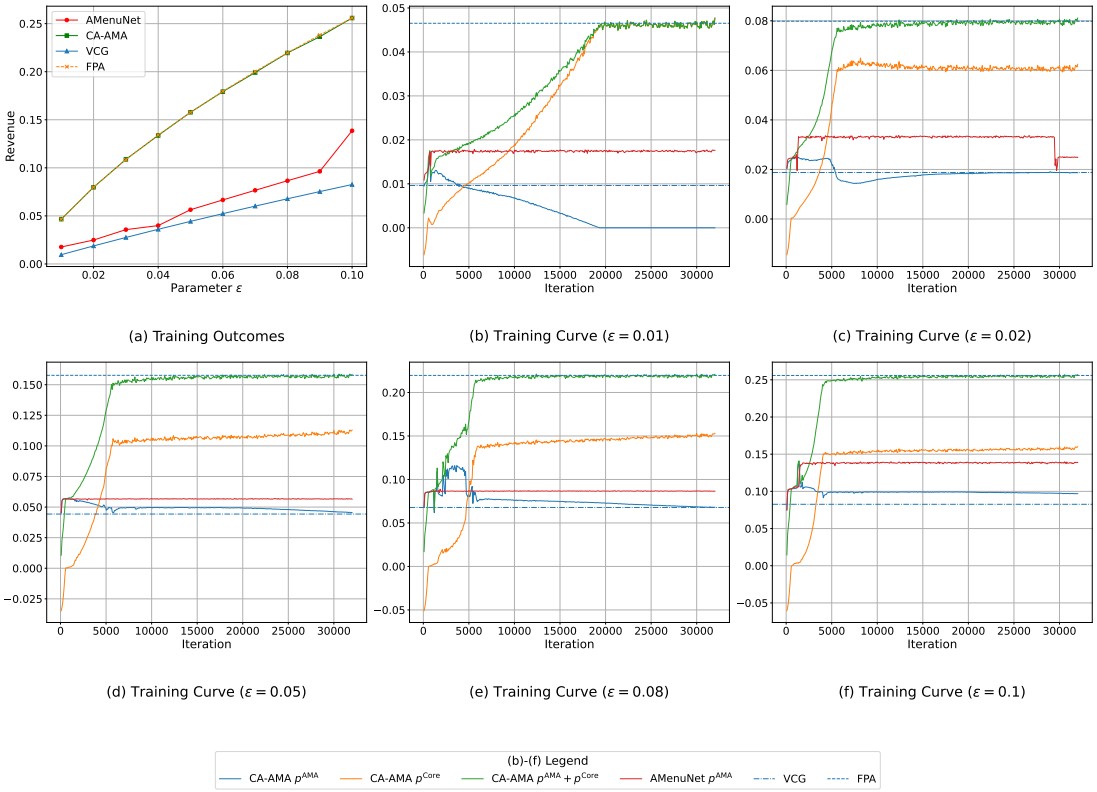

(a) Training Outcomes

(b) Training Curve ($\varepsilon = 0.01$)

(c) Training Curve ($\varepsilon = 0.02$)

(d) Training Curve ($\varepsilon = 0.05$)

(e) Training Curve ($\varepsilon = 0.08$)

(f) Training Curve ($\varepsilon = 0.1$)

(b)-(f) Legend

CA-AMA $p^{\text{AMA}}$ — CA-AMA $p^{\text{Core}}$ — CA-AMA $p^{\text{AMA}} + p^{\text{Core}}$ — AMenuNet $p^{\text{AMA}}$ --- VCG ----- FPA

*Figure 4.* The revenue results and training curves of CA-AMA and Randomized AMA (implemented by AMenuNet (Duan et al., 2023)) in auctions with the first bidder's valuation $v_1$ following equal revenue distribution on $[\epsilon, 1]$ and the second bidder's valuation $v_2 = \frac{\epsilon}{1-\epsilon}(1 - v_1)$. As the $\text{Regret}_{\text{IR}}$ in all cases is less than $1e - 5$, it is not plotted in the figure.

tivariate normal value distributions, as described in detail in Section 5. We evaluate $R_{\text{target}}$ for values in the set $\{0.05, 0.02, 0.01, 0.005, 0.002, 0.001, 0.0005, 0.0001\}$. Figure 5 presents the average revenue and the achieved IR regret over 5 independent test runs for CA-AMA at each target regret level. For comparison, the revenue achieved by Randomized AMA, VCG, and Item-CAN is also included.

Firstly, we observe that after training, the achieved IR regret for CA-AMA is consistently close to the specified target value, even for very small targets like $R_{\text{target}} = 0.0001$. This demonstrates the effectiveness of our training algorithm in steering the mechanism towards a desired level of IR compliance, mitigating the significant IR violations that can occur with standard AMA approaches. Secondly, as $R_{\text{target}}$ approaches 0, the revenue obtained by CA-AMA tends to decrease. Nevertheless, CA-AMA consistently yields higher average revenue than Randomized AMA across all tested target regret levels.

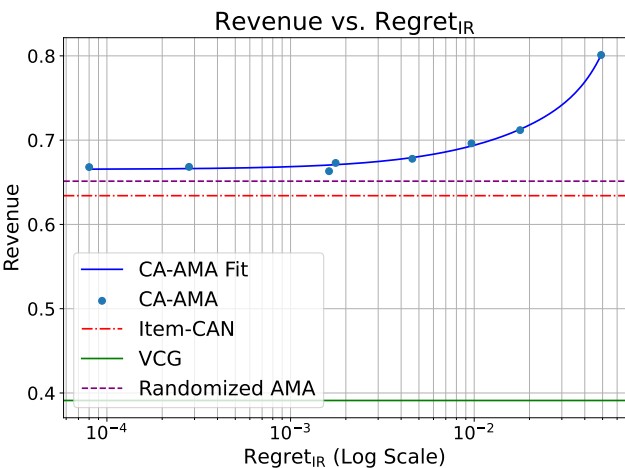

*Figure 5.* Average revenue vs. achieved IR regret for the optimized CA-AMA under different target IR regret ($R_{\text{target}}$). Results are averaged over 5 test runs in a 2-bidder, 2-item auction setting with irregular multivariate normal value distributions. Revenue obtained by Randomized AMA, VCG, and Item-CAN is included for comparison.

