# OpenReview forum: "Enhancing Affine Maximizer Auctions with Correlation-Aware Payment"
_ICML.cc/2026/Conference — ICML 2026 regular_

### Official Review · Reviewer_d62A · 2026-03-05

**Soundness:** 4
**Presentation:** 4
**Significance:** 4
**Originality:** 4
**Overall Recommendation:** 6
**Confidence:** 5

**Summary:**

This paper proposes Correlation-Aware AMA (CA-AMA), which augments the classical Affine Maximizer Auction with a correlation-aware payment term. The goal is to enhance revenue expressiveness under correlated valuation distributions while preserving DSIC.

**Compliance With Llm Reviewing Policy:**

Affirmed.

**Final Justification:**

We are very grateful for the author's reply. Based on the original manuscript and the author's reply, the above is our final score for this article.

**Key Questions For Authors:**

1. Although the paper provides a strict separation theorem in the single-item case, the multi-item setting is supported only by empirical results. There is no corresponding theoretical characterization. It remains unclear whether CA-AMA maintains a strict expressiveness advantage under general multi-item correlated distributions, or whether the advantage is limited to specific structures. This creates a gap between theory and experiments in the multi-item section.

2. The paper compares IR-regret with IC-regret and emphasizes the computational efficiency and flexibility of the former. However, the two regret terms correspond to fundamentally different constraints and are therefore not directly comparable at the mechanism level. IC-regret enforces incentive compatibility, which is central to implementability, whereas IR-regret only controls individual rationality violations. Since CA-AMA guarantees DSIC by construction and does not optimize over IC constraints during training, its computational advantage largely stems from not needing to handle IC optimization, rather than from a more efficient solution to an equivalent problem.

**Limitations:**

yes

**Strengths And Weaknesses:**

Strengths
The article is rigorously written, highly innovative, and fluently expressed, effectively addressing the current Correlation-Aware Payment problem. Specifically
1. The paper clearly identifies that classical Affine Maximizer Auctions (AMA) suffer from limited expressiveness in correlated valuation settings, due to the monotonicity constraints imposed by VCG-style payments. The authors not only provide intuitive examples but also rigorously formalize this limitation. This structural boundary of the AMA family has not been systematically characterized in prior work, and the paper provides important clarification of its theoretical limits.

2. CA-AMA enhances revenue expressiveness by introducing a correlation-aware payment term that depends only on other bidders’ reports and not on the bidder’s own report. This modification preserves dominant-strategy incentive compatibility while expanding the mechanism’s revenue capabilities. The design follows the envelope principle and is structurally clean and easy to verify.

3. The paper proves that the proposed network satisfies DSIC. In the single-item setting, it establishes a strict separation result between deterministic AMA and CA-AMA. In addition, it analyzes the continuity and learnability of the optimal correlation-aware payment term. The authors prove continuity with respect to AMA parameters and, under suitable conditions, with respect to other bidders’ valuations. They further provide a generalization bound for IR-regret, showing that violations of individual rationality can be statistically controlled under finite samples. Overall, the paper provides fairly comprehensive theoretical support for the expressiveness, stability, and generalization properties of CA-AMA.

Weaknesses

1. Although the paper provides a strict separation theorem in the single-item case, the multi-item setting is supported only by empirical results. There is no corresponding theoretical characterization. It remains unclear whether CA-AMA maintains a strict expressiveness advantage under general multi-item correlated distributions, or whether the advantage is limited to specific structures. This creates a gap between theory and experiments in the multi-item section.

2. The paper compares IR-regret with IC-regret and emphasizes the computational efficiency and flexibility of the former. However, the two regret terms correspond to fundamentally different constraints and are therefore not directly comparable at the mechanism level. IC-regret enforces incentive compatibility, which is central to implementability, whereas IR-regret only controls individual rationality violations. Since CA-AMA guarantees DSIC by construction and does not optimize over IC constraints during training, its computational advantage largely stems from not needing to handle IC optimization, rather than from a more efficient solution to an equivalent problem.

3. Some symbols require more detailed explanation, for example in Proposition 3.1 REV^D-AMA_F and REV^S-AMA_F. Clearer notation and definitions would improve readability and precision.

---

> ### Author Rebuttal · Authors · 2026-03-30
>
> We sincerely thank the reviewer for the careful assessment and for the strong accept rating, which recognizes the overall contribution of this paper. We address the concerns as follows.
>
> ---
>
> **Response to Weakness 1 & Key Question 1 (Multi-item theory):**
>
> We would like to add further discussion on the comparison between CA-AMA and AMA in multi-item auctions. Informally, we believe CA-AMA will have a strict advantage over AMA when there is **negative correlation** between bidders' valuations. The intuition is the same as that in our introduction: *a bidder's payment can only be a non-decreasing function of other bidders' valuations* under classic AMA. Since we expect a bidder's payment to increase with its own valuation, classic AMA cannot express this property under negative correlation, which may lead to revenue loss. Therefore, we believe that the distributions used in our experiments (Dirichlet Value Share and Linear Mixture) are not merely corner cases but have representative significance.
>
> Theoretically, the constructive proof of the second part of Theorem 3.1 can be directly applied to multi-item auctions: we can set all bidders to have *identical* valuations across all items following our constructed single-item distribution. For independent valuations, from our perspective, we lean toward the conclusion that "CA-AMA = AMA" for multi-item auctions. However, a rigorous statement and proof remain open problems. As we have spent considerable time attempting this proof, we discuss several theoretical challenges around Line 190. The main difficulty is that characterizing the optimal parameters even for classic AMA is hard, making the comparison truly challenging.
>
> Overall, we believe that further formalization of a fine-grained theoretical expressiveness comparison between CA-AMA and AMA is a valuable future direction.
>
> ---
>
> **Response to Weakness 2 & Key Question 2 (IR-regret vs IC-regret comparison):**
>
> We thank the reviewer for pointing out this specific issue regarding the comparison between IR-regret and IC-regret. We agree that the computational efficiency mainly stems from the structural advantage of AMA itself, rather than from the introduced correlation-aware payment. However, we would like to clarify that our comparison was intended to highlight differences *across methods*: because CA-AMA inherits the DSIC property from AMA, it does not need to compute IC-regret during training, which yields computational efficiency advantages. We will adjust the logic of this comparison accordingly in the revised paper.
>
> ---
>
> **Response to Weakness 3 (Notation):**
>
> Thank you for pointing this out. These two notations come from our defined optimization problem (AMA-OPT) around the left column of Line 146. Specifically:
>
> - $\text{REV}^{S-AMA}_{\mathcal F}$ refers to the optimal revenue for a **randomized AMA** with menu size pre-set to $S$.
> - $\text{REV}^{D-AMA}_{\mathcal F}$ refers to the optimal revenue for a **deterministic AMA**.
>
> We will clarify this definition to improve readability and precision.
>
> ---
>
> We hope these clarifications will address the reviewer's concerns. Thank you again for the thoughtful review.

---

> > ### Author Rebuttal · Reviewer_d62A · 2026-04-01
> >
> > Thank you very much for the author's reply. I have no further questions regarding the current response.

---

> > > ### Author Response · Authors · 2026-04-04
> > >
> > > We thank the reviewer for the follow-up and for maintaining the positive score. The constructive suggestions will definitely help improve the paper's quality.

---

### Official Review · Reviewer_y1qk · 2026-03-12

**Soundness:** 2
**Presentation:** 3
**Significance:** 2
**Originality:** 3
**Overall Recommendation:** 5
**Confidence:** 5

**Summary:**

This paper proposes a relevance-aware affine maximization auction (CA-AMA) to address the poor returns of the classic affine maximization auction (AMA) in the context of bidder valuation relevance distributions. CA-AMA enhances the expressiveness of the payoff rules by introducing a relevance-aware payoff that depends solely on the valuations of other bidders, while preserving dominant strategy incentive compatibility (DSIC).

**Compliance With Llm Reviewing Policy:**

Affirmed.

**Final Justification:**

After reviewing the authors' thorough and convincing rebuttal, I have upgraded my score from Weak Accept to Accept. My main concerns regarding the need for deeper theoretical insight into multi-item settings and the computational overhead at inference were satisfactorily addressed. The authors provided valuable clarifications and additional perspective, reinforcing the paper's originality in leveraging endogenous correlation within the AMA framework and its significance as a practical advancement.

**Key Questions For Authors:**

1. Depth of experimental analysis: The experimental section excellently demonstrates the yield improvement, but the detailed analysis of IR violations could be more in-depth. For example, reporting the proportion of samples with negative utility under different mechanisms, rather than just the average IR regret value.

2. Comparison with broader baselines: GemNet is understandably excluded from comparison due to its implementation complexity. However, a brief qualitative comparison of the conceptual similarities and differences between CA-AMA and mechanisms like GemNet in the discussion would help place this work more accurately within the current research landscape.

**Limitations:**

yes

**Strengths And Weaknesses:**

Strengths:

1. High technical rigor: The paper establishes a complete and solid technical system.

2. Clear and structured expression: The paper follows the standard academic paper structure, with a smooth narrative and clear logical progression.

3. Clear importance and practicality: The paper solves an important problem in the field of automated mechanism design that has been identified but not well addressed by the classic AMA framework: designing high-yield and property-guaranteed auctions when bidder valuations are relevant.

Weaknesses:

1. Depth of experimental analysis: The experimental section excellently demonstrates the yield improvement, but the detailed analysis of IR violations could be more in-depth. For example, reporting the proportion of samples with negative utility under different mechanisms, rather than just the average IR regret value.

2. Comparison with broader baselines: GemNet is understandably excluded from comparison due to its implementation complexity. However, a brief qualitative comparison of the conceptual similarities and differences between CA-AMA and mechanisms like GemNet in the discussion would help place this work more accurately within the current research landscape.

---

> ### Author Rebuttal · Authors · 2026-03-30
>
> **We sincerely thank the reviewer for the constructive feedback and for recognizing the technical strengths of our paper.** We address your specific concerns below.
>
> ---
>
> **Response to Weakness 1 & Key Question 1 (Depth of IR violation analysis):**
>
> We appreciate this suggestion and agree that a deeper analysis of IR violations strengthens the paper. To address this, we have conducted additional experiments measuring the exact proportion of negative utility cases across the different learned mechanisms. The first four rows correspond to the **Dirichlet Value Share ($\alpha$ = 0.5)** and the last four rows correspond to the **Dirichlet Value Share ($\alpha$ = 2.0)**.
>
> | | Fraction of Negative Utility | Regret (q25) | Regret (q50) | Regret (q75) | Regret (q90) | Regret (q95)  |
> |------|-------------------------------|--------------|--------------|--------------|--------------|---------------|
> | 2x2  | 0.0200 | 0.0055  | 0.0130  | 0.0362  | 0.0639  | 0.0805   |
> | 2x5  | 0.0393 | 0.0147  | 0.0396  | 0.0845  | 0.1359  | 0.1745   |
> | 3x10 | 0.0120 | 0.0206  | 0.0542  | 0.1148  | 0.1954  | 0.2481   |
> | 5x5  | 0.0047 | 0.0094  | 0.0248  | 0.0516  | 0.0875  | 0.1144   |
> | 2x2  | 0.0129 | 0.0092  | 0.0227  | 0.0424  | 0.0656  | 0.0831   |
> | 2x5  | 0.0239 | 0.0145  | 0.0357  | 0.0775  | 0.1258  | 0.1598   |
> | 3x10 | 0.0046 | 0.0142  | 0.0402  | 0.0913  | 0.1525  | 0.1941   |
> | 5x5  | 0.0081 | 0.0050  | 0.0120  | 0.0237  | 0.0362  | 0.0472   |
>
> Specifically, we found that across all tested distributions, the opt-out rate for CA-AMA remained below 5%, confirming that the learned IR regret is tightly bounded. We will add the column **"Fraction of Negative Utility"** to Table 1.
>
> ---
>
>
> **Response to Weakness 2 & Key Question 2 (Comparison with broader baselines and GemNet):**
>
> We agree that a more in-depth discussion comparing CA-AMA with fully expressive baselines like GemNet helps clarify the motivation and positioning of our work. Below, we provide further discussion on the comparison between CA-AMA and other representative works.
>
> CA-AMA can be interpreted as a specialized menu-based mechanism. When the ex-post opt-out option is introduced, CA-AMA essentially offers each bidder a menu with exactly **two options**: (1) the allocation and payment derived by the CA-AMA network, or (2) zero allocation for zero payment. While general menu-based mechanisms (e.g., GemNet) can theoretically express any DSIC and IR mechanism, CA-AMA's simplified two-option structure means it may not capture the absolutely optimal mechanism across all possible distributions.
>
> However, we emphasize two crucial practical advantages that CA-AMA holds over mechanisms with full theoretical expressiveness (e.g., RegretNet and GemNet):
>
> 1. **Versus RegretNet:** Because CA-AMA intrinsically guarantees DSIC, it entirely avoids the computationally expensive process of calculating IC-regret during training.
> 2. **Versus GemNet:** CA-AMA naturally satisfies strict allocation feasibility without requiring the complex integer programming or post-processing steps necessary in GemNet, making it significantly easier to optimize and faster at runtime.
>
> **Conclusion:** While CA-AMA trades off full theoretical expressiveness, it consistently achieves highly competitive empirical revenue. Consequently, we view CA-AMA as occupying a valuable position on the Pareto front between theoretical expressiveness, strict incentive guarantees, and empirical performance.
>
> We will integrate the above discussion into the Related Work and Appendix sections in the revised paper.
>
> ---
>
> We hope these clarifications will address the reviewer's concerns. Thank you again for the thoughtful review.

---

> > ### Author Rebuttal · Reviewer_y1qk · 2026-04-02
> >
> > Thank you for your response. We appreciate that our concerns have been adequately addressed.

---

> > > ### Author Response · Authors · 2026-04-04
> > >
> > > We thank the reviewer for the follow-up and for raising the score. The constructive suggestions will definitely help improve the paper's quality.

---

### Official Review · Reviewer_eaDH · 2026-03-13

**Soundness:** 3
**Presentation:** 4
**Significance:** 3
**Originality:** 3
**Overall Recommendation:** 4
**Confidence:** 4

**Summary:**

The paper proposes Correlation-Aware Affine Maximizer Auctions (CA-AMA), a modification of affine maximizer auctions designed to better exploit correlations between bidders’ values. Standard AMAs guarantee dominant-strategy incentive compatibility (DSIC) since they are a VCG-style mechanism; as well as individual rationality (IR), but their payment rules are not expressive enough to capture certain revenue-optimal mechanisms when bidders’ valuations are correlated.

To address the setting of correlated bidders' valuations, CA-AMA augments vanilla AMAs with an additional payment that only depends on other bidders' reports and accounts for correlations. Because this additional payment term does not depend on the bidder’s own bid, the mechanism remains DSIC. However, IR may not hold, and must be enforced through a two-stage neural network that minimizes revenue loss plus IR regret, as well as some post processing for IR.

The main theoretical result is that under correlated distributions, deterministic AMAs can be arbitrarily suboptimal, while CA-AMA can achieve optimal revenue in single-item auctions with bidder-independent distributions. The authors also characterize the optimal correlation-aware payment and prove their continuity and generalization bounds that justify learning it with neural networks. Empirical results across several correlated multi-item environments show that CA-AMA achieves higher revenue than randomized AMA, VCG, and item-wise CAN baselines.

**Compliance With Llm Reviewing Policy:**

Affirmed.

**Key Questions For Authors:**

1. Do the correlation-aware payments exhibit desirable properties like monotonicity or linear dependence in other bidders' reports? Can the authors comment on whether such properties can be enforced in training?

2. Can the revenue loss due to bidders opting out in the final stage for IR be theoretically bounded?

**Limitations:**

I think the authors should expand on the limitations and the impact statement, since mechanisms that exploit correlations between bidders information could lead to discriminatory pricing or unwanted strategic behavior.

**Strengths And Weaknesses:**

**Strengths:**

The paper is very well-written. The problem of learning correlation-aware payments for AMAs with correlated bidder valuation is well-motivated; moreover the separation result between correlation-aware AMAs and AMAs justifies the proposed idea. The proposed empirical solution is reasonable and does well in practice.

**Weaknesses:**

The theoretical contributions of the paper are somewhat limited: the separation result applies only to single item setting, and the results from Section 4 try to justify the learnability of the correlation-aware payments but do not really comment on the quality of the learned payment in terms of approximating the optimal revenue, which would have been a valuable contribution. Moreover, it does not seem like some desirable structural properties of the correlation-aware payments (e.g. monotonicity, linear dependence) are not guaranteed. Finally, although IR is trained for, it should also be enforced ex-post by allowing bidders to opt out if they receive negative utility. This does preserve DSIC, but it is not clear if this step can lead to a large loss in revenue.

---

> ### Author Rebuttal · Authors · 2026-03-30
>
> **We sincerely thank the reviewer for the detailed, constructive feedback and for recognizing the strengths and potential impact of our work.** We address your specific concerns below.
>
> ---
>
> **Response to Weakness 1 (Theoretical contributions):**
>
> - **Multi-item extensions:** The second part of Theorem 3.3 (CA-AMA) directly extends to multi-item settings: if bidders have *identical* valuations across all items following our constructed single-item distribution, deterministic AMA performs arbitrarily poorly and randomized AMA remains suboptimal, whereas CA-AMA achieves optimality. For independent valuations, proving "CA-AMA = AMA" remains a conjectured future direction due to the theoretical challenges discussed (Line 190).
>
> - **Learned payments:** Standard statistical learning techniques can bound empirical vs. true distribution performance (similar to Theorem 4.2). Bounding the exact distance to the true optimal mechanism remains an open problem, since characterizing the true optimal AMA for general distributions is unsolved. Thus, we focus on empirical validation, showing near-optimal performance in solvable special cases (Fig. 2).
>
> ---
>
> **Response to Weakness 2 & Key Question 1 (Structural properties of correlation-aware payments):**
>
> As derived in the paper (Line 265, right), the target form of the correlation-aware payment is:
>
> $$p_i^{\text{OPT-core}}(V_{-i}) = \inf_{v_i \in \text{supp}(\mathcal F_i(V_{-i}))} u_i^{\text{AMA}}((v_i, V_{-i}); \mathcal A, w, \lambda).$$
>
> For a fixed menu $\mathcal A$ of size $K$, the utility $u_i^{\text{AMA}}$ is
>
> $$u_i^{\text{AMA}} = \max\_{k \in [K]} \left(\sum\_{j=1}^N w_j v_j \cdot (A_k)_j + \lambda_k\right) - \max\_{k \in [K]} \left(\sum\_{j=1, j\neq i}^N w_j v_j \cdot (A_k)_j + \lambda_k\right).$$
>
> Thus, **the correlation-aware payment is a difference of two maxima of linear functions**. This structure characterizes the target correlation-aware function in theory.
>
> In our experiments, the correlation-aware payment is modeled as a neural network without specific structural constraints, as this is easy to implement and demonstrates good empirical performance. We will add this clarification to the revised paper.
>
> ---
>
> **Response to Weakness 3 & Key Question 2 (Theoretical guarantee on revenue loss due to opt-out):**
>
> The reviewer raises a highly perceptive point. Because an opt-out requires a bidder to forfeit their entire payment rather than just the violated amount, a mechanism with frequent but microscopic IR violations could theoretically suffer a disproportionately large revenue loss.
>
> To address this, we make a preliminary analysis of the relationship between revenue loss and IR regret. Let the IR regret $r_i$ be defined as $r_i = \max \\{0, p_i - v_i \cdot g_i \\} $, where $g_i$ is the allocation vector for bidder $i$. If a bidder opts out, we have $r_i = p_i - v_i \cdot g_i$, i.e., $p_i = r_i + v_i \cdot g_i$.
>
> The expected total revenue loss $\Delta \text{Rev}$ from all opt-outs is:
>
> $$\Delta \text{Rev} = \mathbb{E} \left[ \sum_{i=1}^n p_i \cdot \mathbb{I}(u_i < 0) \right] = \mathbb{E} \left[ \sum_{i=1}^n (r_i + v_i \cdot g_i) \cdot \mathbb{I}(u_i < 0) \right].$$
>
> Assuming valuations are normalized within $[0, 1]$ across $m$ items, the maximum possible allocated value $v_i \cdot g_i$ is bounded by $m$. This yields the bound:
>
> $$\Delta \text{Rev} \le \mathbb{E}[\text{Regret-IR}] + m \cdot n \cdot \mathbb{P}(\text{opt-out}).$$
>
> Thus, revenue loss is strictly bounded by the expected IR regret plus the maximum value of forfeited items. When $\mathbb{P}(\text{opt-out})$ is driven near zero in practice, the revenue loss can be made small.
>
> ---
>
> **Response to Limitations (Expand limitations and impact statement):**
>
> We thank the reviewer for highlighting these ethical and practical considerations. We will add the following Broader Impact section:
>
> > While CA-AMA advances automated mechanism design theoretically, its practical deployment in environments like ad auctions carries significant economic implications. By leveraging peer information to set personalized, correlation-aware payments ($p^{\text{core}}$), the mechanism inherently engages in algorithmic price discrimination. Bidders with identical true valuations may face different effective reserve prices depending on the inferred behavior of others. Furthermore, while CA-AMA maintains DSIC for non-cooperative individuals, it may inadvertently encourage unwanted strategic behavior in highly correlated environments. Bidders aware of these dynamics might be incentivized to form collusive rings or employ artificial bid-shading to manipulate payment terms, highlighting a crucial trade-off between revenue maximization and market fairness.
>
> ---
>
> We hope these clarifications will address the reviewer's concerns. Thank you again for the thoughtful review.

---

> > ### Author Rebuttal · Reviewer_eaDH · 2026-04-04
> >
> > Thanks for the reponse. I'll remain with my (already positive) score.

---

> > > ### Author Response · Authors · 2026-04-04
> > >
> > > We thank the reviewer for the follow-up and for maintaining the positive score. The constructive suggestions will definitely help improve the paper's quality.

---

### Official Review · Reviewer_cvXf · 2026-03-13

**Soundness:** 3
**Presentation:** 3
**Significance:** 2
**Originality:** 2
**Overall Recommendation:** 4
**Confidence:** 4

**Summary:**

This paper addresses the revenue limitations of traditional Affine Maximizer Auctions (AMAs) in markets with correlated bidder valuations. Standard AMAs, widely used in differentiable economics for their inherent dominant-strategy incentive compatibility (DSIC) and individual rationality (IR), face severe revenue bottlenecks under correlated valuations due to their restrictive VCG-style payment rules. To overcome this, the authors propose the Correlation-Aware AMA (CA-AMA) framework, which augments the standard mechanism with neural-network-parameterized payment terms for each bidder that depend on other bidders’ valuations. Since this additional charge is independent of a bidder’s own report, CA-AMA strictly preserves the DSIC property while drastically enhancing expressive power. The authors formulate the mechanism design as an IR-constrained revenue maximization problem via a novel two-stage deep learning algorithm with theoretical generalization bounds on IR violations. Moreover, the authors have provided several experiments to showcase that CA-AMA achieves near-optimal revenue in complex correlated environments where classic AMAs perform arbitrarily poorly.

**Compliance With Llm Reviewing Policy:**

Affirmed.

**Final Justification:**

My concerns have been adequately addressed.

**Key Questions For Authors:**

I suggest the authors further clarify the primary motivation driving their framework. It is currently somewhat ambiguous whether the value-correlated setting was introduced specifically to facilitate the relaxation of the IR constraint, or if the need to relax IR was discovered a posteriori as a technical requirement to solve the value-correlated scenario.

**Limitations:**

Yes.

**Strengths And Weaknesses:**

Strength:

**Originality**：The originality lies in augmenting the standard AMA framework by adding a neural-network-parameterized payment term $p_i^{Cor}(V_{-i})$ that depends strictly on other bidders' valuations. This specific architectural choice to bypass the expressiveness bottleneck of VCG-style payments is a novel idea in AMD.

**Soundness**:  The construction in Theorem 3.3 clearly and mathematically proves exactly why and how much standard AMAs fail in correlated settings, justifying the need for their proposed mechanism.

**Significance**：Standard AMA-based neural networks (like AMenuNet) are currently the state-of-the-art for designing strict DSIC auctions, but their inability to handle correlated valuations was a known severe limitation. By fixing this, the paper significantly narrows the gap between theoretical optimal mechanisms and implementable, scalable algorithms.

**Presentation**：The paper is adorably clear, well-structured.

Weakness:

In Theorem 4.1, the theoretical continuity of the optimal payment relies heavily on the infimum over the exact support of the conditional distribution $ \text{supp}(\mathcal{F}_i(V_{-i}))$. In any real-world data-driven setting, you only have empirical samples, not the true support boundaries. It is doubtful whether the model can be extended to real-world applications.

The concept of IR regret is introduced based on the premise that AMA can strictly guarantee IC. However, as mentioned in theorem 3.3, in independent scenarios, no additional payment is needed. I think the author needs to clarify their motivation: did this article choose the value-correlated scenario to relax IR, or did the technique for relaxing IR come about after choosing the specific scenario?

---

> ### Author Rebuttal · Authors · 2026-03-30
>
> We sincerely thank the reviewer for the detailed, constructive feedback and for recognizing the originality, soundness, and presentation of our work. We address your specific concerns below.
>
> ---
>
> **Response to Weakness 1: Applicability of Theorem 4.1 in Data-Driven Settings**
>
> We agree with the reviewer that in real-world, data-driven applications, we only have access to empirical samples rather than the exact true support boundaries, $\text{supp}(\mathcal{F}\_i(V\_{-i}))$. We would like to clarify that **Theorem 4.1 is not intended to provide an analytical formula to be computed directly from samples at runtime**. Instead, it serves as the **foundational theoretical justification for employing a neural network**.
>
> Specifically, Theorem 4.1 establishes that the ideal target function for the optimal payment is continuous. This continuity guarantees that a neural network can theoretically learn and express this function. In practice, our framework trains a neural network to learn this function from empirical samples and then generalizes to unseen data. Statistically, when the training sample size is sufficiently large, the empirical distribution densely covers the support, allowing it to approximate the true infimum with high probability.
>
> The significant revenue improvements demonstrated in our experiments confirm that the network successfully approximates this target function from sampled data alone, validating its real-world applicability. We will add a discussion bridging this theoretical continuity and empirical sampling to Section 4.
>
> ---
>
> **Response to Weakness 2 & Key Question: Motivation for IR Regret**
>
> We appreciate the opportunity to clarify the overall motivation of this work and the rationale of our design choices. To directly answer the reviewer's question: **addressing the revenue limitations in value-correlated scenarios without sacrificing the IR property was our primary, original motivation**. The concept of "IR regret" emerged *a posteriori* strictly as a technical requirement for practical optimization.
>
> We can break down our design logic into three steps:
>
> 1. **Primary Motivation:** Standard AMAs suffer severe revenue bottlenecks under bidder-correlated distributions. We proposed the correlation-aware payment term specifically to address this limitation.
>
> 2. **Theoretical Strict IR:** Conceptually, our goal is *not* to relax IR. As defined in our core optimization problem (CA-AMA-OPT, Line 188), the theoretically optimal correlation-aware payment mathematically guarantees strict IR across all distributions. Ideally, we would like to find the exact optimal correlation-aware payment for each bidder, which satisfies the *strict* IR property.
>
> 3. **Practical IR Regret:** The concept of "IR regret" arises from practical neural network optimization. Because we cannot strictly enforce hard IR constraints at every step of gradient descent, we introduced IR regret as a differentiable loss penalty to guide training and as an empirical evaluation metric.
>
> In short, relaxing strict IR via a regret penalty is an operational tool for neural network training. We still expect the fully trained CA-AMA to satisfy strict IR. This is also the reason why we introduce the opt-out choice after the allocation and payment outcome are computed by a CA-AMA, which is used to make sure that each bidder can have a non-negative utility. For clarity, we will explicitly integrate this motivation into the place where IR regret is introduced.
>
> ---
>
> We hope these clarifications can address the reviewer's concerns. Thank you again for the thoughtful review.

---

> > ### Author Rebuttal · Reviewer_cvXf · 2026-04-03
> >
> > My concerns have been adequately addressed. I am increasing my score by 1.

---

> > > ### Author Response · Authors · 2026-04-04
> > >
> > > We thank the reviewer for the follow-up and for raising the score. The constructive suggestions will definitely help improve the paper's quality.

---

### Decision · Program_Chairs · 2026-04-30

**Decision:**

Accept (regular)

**Comment:**

The paper studies the family of affine-maximizer auctions (AMAs, a generalization of VCG) that is DSIC by design, but rather restrictive because of the VCG payment rule's structure. It is well known that VCG and affine maximizer auctions can be quite poor at extracting revenue. In particular for settings with correlated valuations, AMA can perform arbitrarily poorly even in settings where some DSIC, IR mechanisms can extract the entire surplus as revenue. Therefore, the authors add a correlation-aware payment term to each bidder's payment to fix this shortcoming, and proceed to optimize AMAs, which requires optimizing the AMA paramters (namely, the weights and the constant term in the affine maximizer) and also the correlation-aware payment term. The paper designs a two-stage training algorithm with post-processing for this purpose, and experimentally compares the revenue surfaces and runtimes of the correlation aware AMA (CA-AMA) and the plain AMA.

The review team appreciated the fact that the paper seeks to fix a known limitation of a standard paradigm for designing DSIC auctions. Even though the theoretical contributions were perceived to be somewhat thin, the very clearly structured writing along with thorough experimental evaluation makes up for this. Overall this is a good paper for ICML.